

# Seasonal and ENSO-related ocean variability in the Panama Bight

Rafael R. Torres[1], Estefanía Giraldo[2], Cristian Muñoz[2], Ana Caicedo[2], Ismael Hernández-Carrasco[3], Alejandro Orfila[3]

[1]Grupo de Investigación en Geociencias GEO4, Departamento de Física y Geociencias, Universidad del Norte, km 5 vía Puerto Colombia, Barranquilla, Colombia
[2]Dirección General Marítima de Colombia, Centro de Investigaciones Oceanográficas e Hidrográficas del Pacífico (DIMAR-CCCP), Vía El Morro, Capitanía de Puerto San Andrés de Tumaco, Nariño, Colombia
[3]Mediterranean Institute for Advanced Studies, IMEDEA (CSIC-UIB), Esporles 07190, Mallorca, Spain

Correspondence to: Alejandro Orfila (aorfila@imedea.uib-csic.es)

**Abstract.**

In the Panama Bight, a reverse seasonal surface circulation coincides with a strong mean sea level variation, as observed from 27 years of Absolute Dynamic Topography (ADT) and the use of Self-Organizing Maps. From January to April, a cyclonic gyre dominates the basin circulation, forced by the Panama surface wind jet that also produce upwelling, reducing Sea Surface Temperature (SST) and increasing Sea Surface Salinity (SSS), causing an ADT decrease. From June to December, the Choco surface wind jet enhances SST, precipitation and river runoff, which reduces SSS causing an ADT rise, which forces a weak anticyclonic circulation. Interannual variability in the region is strongly affected by ENSO, however this climatic variability does not modify the seasonal circulation patterns in the Panama Bight. On the contrary, ENSO positive (negative) phase increases (decreases) SST and ADT in the Panama Bight, with a mean annual difference of 0.9 °C and 9.1 cm respectively between the two conditions, while its effect in SSS is small. However, as the strong seasonal SST, SSS and ADT ranges are up to 2.2°C, 2.59 gr kg$^{-1}$ and 28.5 cm, the seasonal signal dominates over interannual variations in the Bight.

## 1 Introduction

The Pacific Ocean covers about half of the ocean's Earth surface and thus with ocean-atmosphere coupled processes affecting the entire planet's climate (e.g. Xue et al., 2020). In the Eastern Tropical Pacific (hereinafter ETP), the westerly north and south equatorial currents (NEC – SEC) as well as the easterly north equatorial countercurrent (NECC) conform the main circulation (Kessler, 2006). The Panama Bight is placed in the easternmost side of the ETP, limited by Central and South America, north of the equator (Figure 1). Although the circulation of the large subtropical gyres affects the Panama Bight, due to its sheltered position, local factors dominate its seasonal circulation and ocean-properties (Rodríguez Rubio et al., 2007), with the potential to affect a wider region.

The circulation in the Panama Bight has been described in terms of a reverse oceanic gyre, forced by monsoon-like winds with a cyclonic circulation during the boreal winter and an opposite anticyclonic circulation during the boreal summer




(Devis-Morales et al., 2008; Rodríguez Rubio et al., 2003). However, a weakening of the cyclonic circulation during summer has also been proposed (Dimar, 2020; Kessler, 2006). This is an important issue that needs to be clarified, as an opposite seasonal circulation will affect ocean-atmosphere processes in the Bight such as precipitation, river runoff, the

mixed layer depth, ocean vertical stratification, sea level and coastal dynamics, among others. Besides, these physical factors affect Chlorophyll-a, the phytoplankton growth and biodiversity in the area (Corredor-Acosta et al., 2020).

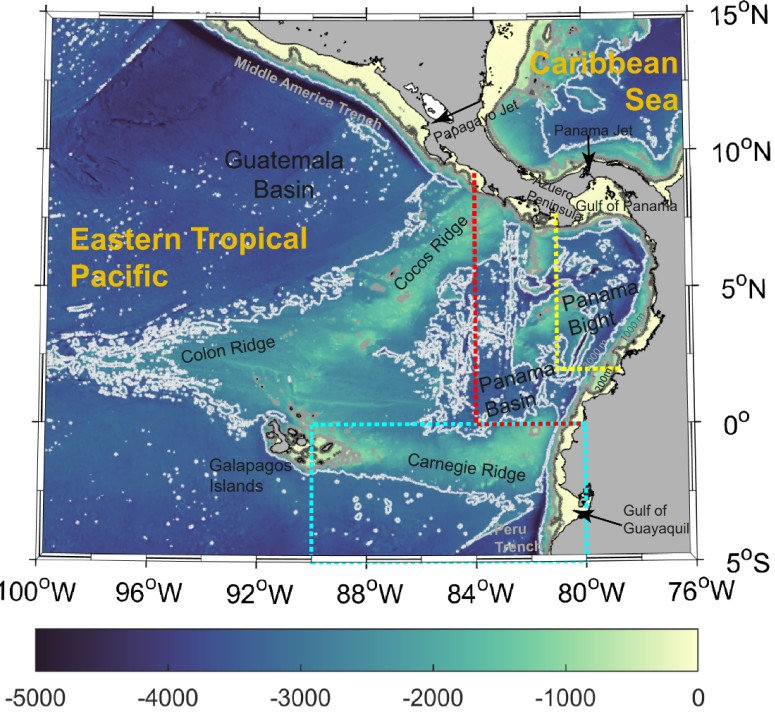

**Figure 1: Map of the Eastern Tropical Pacific (ETP) west to 100°W. The 200, 1000 and 3000 m isobaths are shown in gray. The limits of the Panama Basin/Bight are shown in red/yellow. The cyan box displays the northern area used to compute the Niño 1+2**

El Niño Southern Oscillation – ENSO severely affects ocean and atmospheric dynamics in the ETP at irregular time scales. During the ENSO positive phase (El Niño), the southern trade winds weaken with consequences in the coastal upwelling in the Peruvian coasts. Besides, the western Pacific warm pool migrates towards the east raising the Sea Surface Temperature (SST) and sea level, while deepening the thermocline, which affects the biological productivity and hydrological cycle toward America's coast. By contrast, the ENSO negative phase (La Niña) enhances the normal conditions, lowering the SST

and sea level, while shoaling the thermocline toward the east (Cabarcos et al., 2014; Grados et al., 2018; Kessler, 2006; Trenberth, 1997).



Surface dynamics in the ETP east of 120° W is more complicated than in the Central Pacific, as meridional flow interacts with prevailing zonal currents such as the NECC and SEC. For example, the geostrophic eastward NECC extends towards Central America, merging with the cyclonic circulation around the Costa Rica Dome (Figure A1) with large variability

related to changes in the Papagayo jet, which occurs at weekly time-scales. Besides, the westward SEC is mainly observed in two main lobes, about 3° S and 3° N (Kessler, 2006).

Climate in the Panama Bight is driven by the meridional translation of the Intertropical Convergence Zone (ITCZ) through the year. From December to April, the ITCZ moves southward reaching its southernmost position at ~1°N (Dimar, 2020; Poveda et al., 2006; Villegas et al., 2021). During this season, north trade winds from the Caribbean Sea cross the Panama

Isthmus through orographic gaps, forming the Panama surface wind jet, that affects the Panama Bight ~400 km towards the equator (Rueda Bayona et al., 2007). The stress from this jet forces a cyclonic circulation (counter clockwise) in the Panama Bight, as response to the sea level drop due to divergence of surface waters to the west, causing upwelling. In this region, upwelling forces SST reduction and SSS increase, due to the upwelling of colder and saltier Subtropical Surface Water – STSW (Fiedler and Lavín, 2006). This gyre is formed by the northerly Colombia coastal current (Figure A1), a westward

current in the Panama Gulf, a south-south westerly current at ~81° W (Panama Jet Surface Current) and an easterly current closing the gyre at ~2.5° N (Devis-Morales et al., 2008; Rodríguez Rubio et al., 2003).

Between May and October, the boreal summer, the ITCZ reaches its northernmost position, north of the Panama Isthmus. In this season, the Choco (CHorro del Occidente COlombiano) surface wind jet dominates the area (Poveda and Mesa, 2000). This jet is produced by the south trade winds that cross the equator, rotating towards the northeast due to the Coriolis effect

and the atmospheric pressure and SST meridional gradients. These gradients occurs between the warmer waters in the Panama Bight (lower atmospheric pressure) and the equatorial Cold Tongue (higher atmospheric pressure) formed by coastal and equatorial upwelling and advection of cooler water from the Peru Current (Hastenrath and Lamb, 2004; Zheng et al., 2012). Large SST seasonality in the Cold Tongue also drives the zonal SST gradient across the Equatorial Pacific Ocean, which has important impacts on global climate (Karnauskas et al., 2009).

The Choco jet transports large quantities of moisture inland (3774 $m^3s^{-1}$), forcing a high freshwater contribution during this season, making the Colombian coast one of the most rainiest locations on the Earth (Fiedler and Lavín, 2006; Poveda et al., 2006; Poveda and Mesa, 2000; Tsuchiya and Talley, 1998). The coastline geometry together with the Choco jet produces surface water convergence and thermocline deepening in the Panama Bight, rising SST and ADT, producing an anticyclonic (clockwise) circulation (Devis Morales, 2009; Fiedler and Talley, 2006; Rodríguez Rubio et al., 2003). However, there is

no consensus on this circulation pattern (Kessler, 2006).

Strong El Niño events in the altimetry era, occur during 1997-98 and 2015-16. In this phase, the SST gradient between the equatorial Cold Tongue and the Panama Bight reduces, reducing the Choco wind jet, moisture inshore transport and precipitation, also generating other complex ocean-atmosphere interactions (Fiedler and Talley, 2006; Poveda et al., 2006), which varies locally and depending on each event's characteristics (Dimar, 2020). However, it seems that in the Panama



Bight, negative or weak positive ENSO events do not significantly change the thermal ocean structure; conversely, strong
       positive ENSO increases the ocean's heat content and sea level (Devis Morales, 2009).

       In this context, we review here the seasonal circulation in the Panama Bight from Absolute Dynamic Topography (ADT), as
       it includes the mean ocean currents (Mean Dynamic Topography-MDT), as well as temporal sea level variability (Sea Level
       Anomaly-SLA). Besides, we extend the circulation assessment to the ETP (east of 100° W), to examine the connection

between the Panama Bight and equatorial geostrophic currents. We also study ENSO effects in the Panama Bight sea level,
       and if this forcing has a significant effect on the seasonal circulation. To understand the steric component of sea level
       variations, we also assess SST and Sea Surface Salinity (SSS) variability in the region in seasonal and interannual
       timescales.

## 2 Data and methods

Daily maps of ADT from the global ocean gridded L4 product with a 0.25° x 0.25° resolution for the 1993-2019 period are
       used                                                      (https://resources.marine.copernicus.eu/product-
       detail/SEALEVEL_GLO_PHY_L4_REP_OBSERVATIONS_008_047/INFORMATION, last access November 2022). This
       dataset merges the measurement from the different altimeter missions available. Geostrophic currents from the same product
       are used, which are computed using a 9-point stencil methodology for latitudes outside the ±5° N band (Arbic et al., 2012).

In the equatorial band, the Lagerloef et al. (1999) methodology, introducing the β-plane approximation is used. Besides, we
       download the MDT (CNES-CLS18) and geostrophic currents, corresponding to 1993-2012, which were calculated by
       merging information from altimeter data, GRACE and GOCE gravity field and oceanographic in situ measurements (Mulet
       et al., 2021).

       Monthly SST and SSS fields with the same time span and spatial resolution than ADT were obtained from Copernicus

(https://resources.marine.copernicus.eu/product-
       detail/MULTIOBS_GLO_PHY_TSUV_3D_MYNRT_015_012/INFORMATION, last access November 2022), merging in-
       situ and satellite observations from different projects (Guinehut et al., 2012). We derive TEOS–10 Conservative
       Temperature (Θ) and Absolute Salinity (SA), using the GSW toolbox version 3.06 (McDougall and Barker, 2011).

       The Oceanic Niño Index (ONI) is used to assess ENSO events. Positive/negative ENSO events are identified  by a five

consecutive 3-month running SST mean anomalies computed in the Niño 3.4 region (5°N-5°S; 170°W-120°W), that are
       above/below a threshold of +0.5°C/-0.5°C. Anomalies are computed from 30-year periods, which change every 5 years to
       account for ocean global warming. Monthly series of SST anomalies based on ERSST.v5 product (Huang et al., 2017), were
       downloaded from https://origin.cpc.ncep.noaa.gov/data/indices/ (last access November 2022). SST anomalies are referenced
       to the 1991-2020 period.

To find the months in the 1993-2019 period in which a positive or negative ENSO phase occurred, we use the SST anomaly
       series from the Niño 1+2 (0°-10°S; 90°W-80°W) and Niño 3 (5°N-5°S; 150°W-90°W) oceanic regions, following the same





methodology used by ONI. The former region is used as an indication of ENSO's local effect in the Panama Bight, as this region covers the equatorial Cold Tongue, whose SST variations affect the Choco surface wind jet. The El Niño 3 region is used to assess if results stand with an ENSO index representative of the central equatorial Pacific. First comparison between

indices show that El Niño 1+2 has a strong response to La Niña events, while El Niño 3 is much less responsive to continental influences (Hanley et al., 2003).

We use two different methodological approaches to assess the regional ocean dynamics. First, all months are classified in one of the three ENSO-related conditions, normal, positive and negative (Table 1 and Figure 5f). When El Niño 1+2 is used, normal conditions are the most frequent (58.9% of occurrence). For positive and negative conditions, all monthly means are

computed using between 2 and 8 values, with negative ENSO being more frequent (25.6%) than positive ENSO condition (15.4%). When the El Niño 3 region is used to classify the ENSO conditions, all monthly means are computed using between 3 and 9 values. The frequency of occurrence is in the same order as with El Niño 1+2, with 60.2%, 25.3% and 14.5%, respectively (Figure A3f).

**Table 1. Niño 1+2 distribution of months for the three conditions: normal, El Niño (positive ENSO) and La Niña (negative ENSO). In bold number of months used to assess seasonal differences.**

| Condition | J | F | **M** | A | M | J | J | A | S | O | **N** | D | Tot |
|-----------|----|----|-------|----|----|----|----|----|----|----|-------|----|-----|
| Normal | 18 | 21 | **18** | 15 | 14 | 14 | 14 | 14 | 15 | 16 | **15** | 17 | 191 |
| El Niño | 2 | 2 | **3** | 5 | 6 | 5 | 5 | 6 | 5 | 4 | **4** | 3 | 50 |
| La Niña | 7 | 4 | **6** | 7 | 7 | 8 | 8 | 7 | 7 | 7 | **8** | 7 | 83 |
| Total | 27 | 27 | 27 | 27 | 27 | 27 | 27 | 27 | 27 | 27 | 27 | 27 | 324 |

Computing monthly means under the three ENSO-related conditions, based on the Niño 1+2 SST index, assesses seasonal ADT, SST and SSS spatial anomalies, as well as geostrophic currents in the ETP and Panama Bight. For example, from the

27 available values in March (Table 1), 18 are used to compute the March normal condition, three for the positive and six for the negative ENSO conditions. Anomalies are computed by subtracting to the monthly data the 1993-2019 spatial mean using all data in the ETP (66.0 cm, 26.6 °C and 33.8 gr kg$^{-1}$).

Regional-averaged time series are obtained for the ETP, the Panama Bight and the Cold Tongue for the three ENSO-related conditions. Anomalies are computed by subtracting the 1993-2019 spatial mean respectively. Here, the ETP is defined as

nodes between 5° S to 15° N and 76°-100° W. The Panama Bight limits are 1.875°-9.125° N and 81.125°-77.125° W. These limits were selected to assess the particular local dynamics, which differ from the rest of the ETP. The Cold Tongue region is placed between 1.125°-5.125° S and 81.125°-88.375° W. Note that the Panama Bight and Cold Tongue regions have the same number of nodes (Figure 4).

The second methodological approach uses Self-Organizing Maps (SOM) in order to confirm previous results in the ETP.

SOM is a statistical tool used to compress the information contained in a large amount of data into one single set of maps





(Kohonen, 1982), reducing the high-dimensional feature space of input data to a lower dimensional network of units called neurons. SOM analysis has been used in the oceanography context in several studies (Hernández-Carrasco and Orfila, 2018; Liu et al., 2006; Orfila et al., 2021). Learning processes are carried out by an interactive presentation of the input data to a preselected neuronal network, which is modified during the iterative process. Each unit is represented by a weight vector

with a number of components equal to the dimension of the input data. During each iteration, the neuron whose weight vector is the closest to the presented sample input data vector, called Best-Matching Unit (BMU), is updated together with its topological neighbours towards the input sample. When the probability density function of the input data is approximated by SOM, and each unit is associated with that reference pattern that has a number of components equal to the number of variables in the data set, the training process finishes. The size of the neural network is an important parameter to take into

account to maximize the quality of the SOM analysis. The determination of the size of the neural network is empirical and somehow subjective (Morales-Márquez et al., 2021).

We chose the number of neurons of the network after testing several sizes of the map to check that the cluster structures are shown with sufficient resolution and statistical accuracy. In our case, we have selected for both the temporal and spatial patterns a 3x2-map (6-neurons) configuration, using monthly data from 1993 to 2019 for the ADT, the zonal (U) and

meridional (V) geostrophic velocities in the entire ETP region. The trend, annual and semi-annual cycles were estimated by use of linear regression fitted to the monthly time series obtained from the temporal SOM. Errors were estimated at the 95% confidence level. The percentage of explained variance was calculated from the ratio of the residual variance over the variance of the original series after subtracting the mean and trend of the time series, therefore only accounting for the seasonal cycle. A residual time series was obtained after removing the trend and seasonal cycle. We assessed a causal

relationship between the monthly residuals and ENOS, using the Niño 3 SST time series, evaluating their correlation coefficient with a significance level of $p < 0.01$. We used El Niño 3 since SOM was performed with data from the entire ETP. In the case of the spatial analysis the evolution of a particular pattern is provided by the BMU for each sample while in the temporal domain the analysis of the neurons provides temporal patterns and the BMU is used to localize in space the temporal variability, identifying regions of similar co-variability patterns.

**3 Results and discussion**

To assess the seasonal circulation in the Panama Bight and its interannual variability, we first analyze the mean circulation from the MDT and the SOM in the temporal domain. As a second step, in Section 3.2 we describe the seasonal circulation based on normal months where ENSO conditions were not dominant. We further assess the circulation during ENSO positive and negative conditions in Section 3.3, in order to observe differences from the normal circulation patterns. In both

the seasonal and interannual circulation assessment, we analyze the relation between sea level variations and steric changes associated to SST and SSS variability. In Section 3.4 we analyze regionally averaged ADT, SST, SSS and current speed time series, to describe their seasonal behavior under normal, positive and negative ENSO conditions. Finally, in Section 3.5



SOM is used to verify the previously described circulation patterns. In all the sections, we first assess the ETP (east of 100° W), in order to contextualize the Panama Bight dynamics, which is described in more detail.

**3.1 Mean circulation in the Panama Bight from MDT and mean temporal SOM**

Geostrophic currents associated with the MDT (the averaged difference between the mean sea surface and the geoid for the 1993-2012 period) in the ETP displays the SEC as a strong westwards current between 0°- 4°N, distinguishable west of ~85° W (Figure A1a). The SEC results from geostrophic currents produced by positive MDT anomaly in the 3° to 5° N band, and a negative MDT anomaly south of 0.5° N, producing a ~15 cm meridional sea level gradient. The NECC can be identified as

an eastward current, less intense than SEC, between 5° to 7° N reaching ~90° W, from where it starts a counter clockwise rotation around the Costa Rica Dome, which corresponds to a MDT bowl (upwelling), forced by the Papagayo surface wind jet (Figure A1a). East of 100° W, this circulation responds to a MDT gradient between the positive anomaly in the 3° to 5° N band, and the MDT negative anomaly at ~9° N. These circulation patterns coincide with the description given by Kessler (2006).

Mean circulation in the Panama Basin (Figure A1b) differs from the predominant zonal circulation at the same latitudinal band west of ~84° W. In this area, the Panama surface wind jet produces a MDT bowl (upwelling) in the Panama Gulf. The most representative mean circulation feature is a counter clockwise rotation that dominates the northern part of the Panama Bight (east of ~81° W). This circulation shows a strong northerly coastal current known as the Colombia Coastal Current (2.5-7.5°N), a cyclonic current around the Panama Gulf, and the south-westerly Panama Jet Surface Current (Devis-Morales

et al., 2008). The cyclonic rotational closes with weaker eastward currents at ~4.5° N. Therefore, the mean circulation in the region shown by MDT coincides with the dominant circulation at the beginning of the year when the Panama wind jet affects the Panama Bight. Note that the Panama Jet extends southwards, connecting with the SEC at ~2.5° N - 82° W. The mean temporal SOM circulation coincides with the MDT circulation in the ETP, except for the Panama Bight (Figure A1d), where a clockwise circulation pattern is observed south of ~6°N and circulation in the Panama Gulf is eastward.

**3.2 Seasonal circulation in the Panama Bight under normal (no ENSO) conditions**

We assess monthly ADT and associated geostrophic circulation, based on normal ENSO conditions (Table 1), first in the ETP east to 100° W, and later in the Panama Bight. Based on the annual observed behaviour (Figure A2), we show results from two representative months (Figure 2b and e). March represents the circulation from January to April when the Panama wind jet dominates the basin's dynamics. November represents the circulation from June to December, when the Choco wind

jet is dominant.

In the ETP positive ADT anomalies are observed from June to December in the band between 1° to 7° N, extending toward the Colombian coast (Figure 2e). In this area, a weak anticyclonic circulation dominates the Panama Bight east of ~84°W, in response to the positive ADT anomaly (Figure 3f). This circulation shows northwards current around 80-81° W with speeds over 20 cm s$^{-1}$ (Figure 3h). An easterly current dominates the Gulf of Panama, turning into a southerly coastal current along



the Colombian coast, which can be distinguished north of ~4° N, when it reaches the latitude of the San Juan River mouth, from where the anticyclone circulation closes with a westerly current. In this season, the Choco wind jet forces convergence of warm water toward the coast (Figure 4e). Besides, salinity reduces due to the increase of precipitation and river outflow (Figure 4k), reducing surface density and raising ADT in the Panama Bight (black line in Figure 5a, c and d), forcing the relatively weaker anticyclonic circulation. During this season, a clear meridional gradient in the ETP separates cold and

saltier waters to the south from warm and fresh waters to the north. Besides, upwelling forced by the Panama and Papagayo surface wind jets is weak as a small signature can be distinguished in SST and SSS.

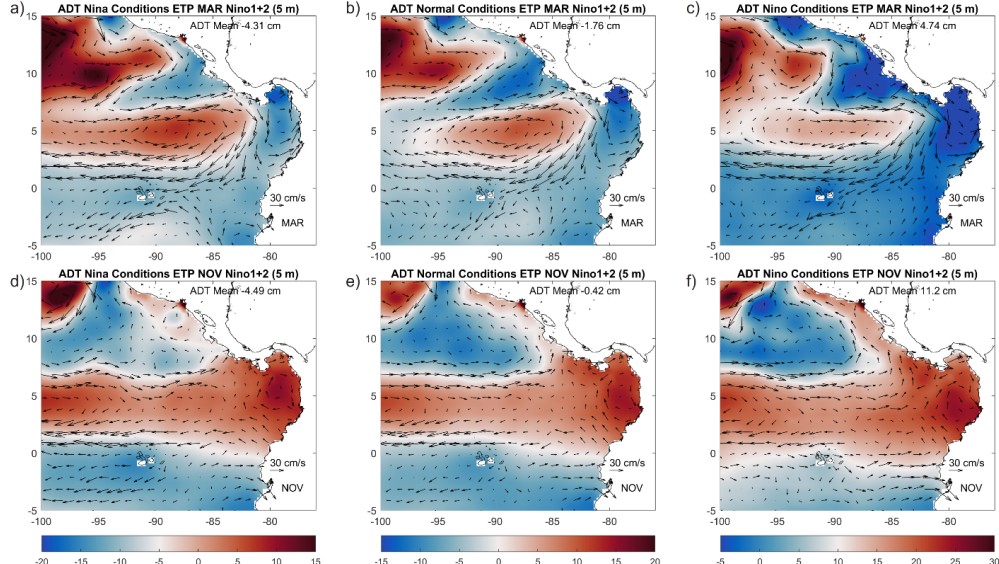

**Figure 2. Monthly average of ADT anomalies (color scale in cm) and associated geostrophic circulation (vectors) for the Eastern Tropical Pacific – ETP. Seasonal changes are indicated within two months: March (a,b,c) and November (d,e,f). ENSO-related**
**conditions (based on Niño 1+2) shown are: normal (b,e), negative ENSO (a,d) and positive ENSO (c,f). Each condition has a different color range to highlight the ADT regional gradients, responsible for the geostrophic circulation. ADT Anomalies are computed subtracting the ETP regional mean during the 1993-2019 period (66.0 cm). The regional average of the ADT anomalies for each month is shown in the upper right of each panel, which coincides with time series values in Figure A3a.**

From January to April, the Papagayo and Panama wind jets strength due to the north trade winds intensification in the
Caribbean. The circulation in the Costa Rica Dome (negative ADT Anomaly) extends south-westwards (Figure 2b), weakening the positive ADT anomaly in the 1°-7° N band, clearly seen in the other season. As a consequence, an anticyclonic circulation (positive ADT anomaly) appears between the Costa Rica Dome and Panama Bight cyclonic circulations. Besides, colder and saltier surface waters also indicate the upwelling intensification forced by these two surface wind jets. In this season the SST and SSS meridional gradients between the Panama Bight and the Cold Tongue weakens




when compared to the other season (Figure 4b and h). Note a warming during these months of the Cold Tongue, observed at

the south of the ETP, with small seasonal changes in its salinity, except for a northern coastal migration.

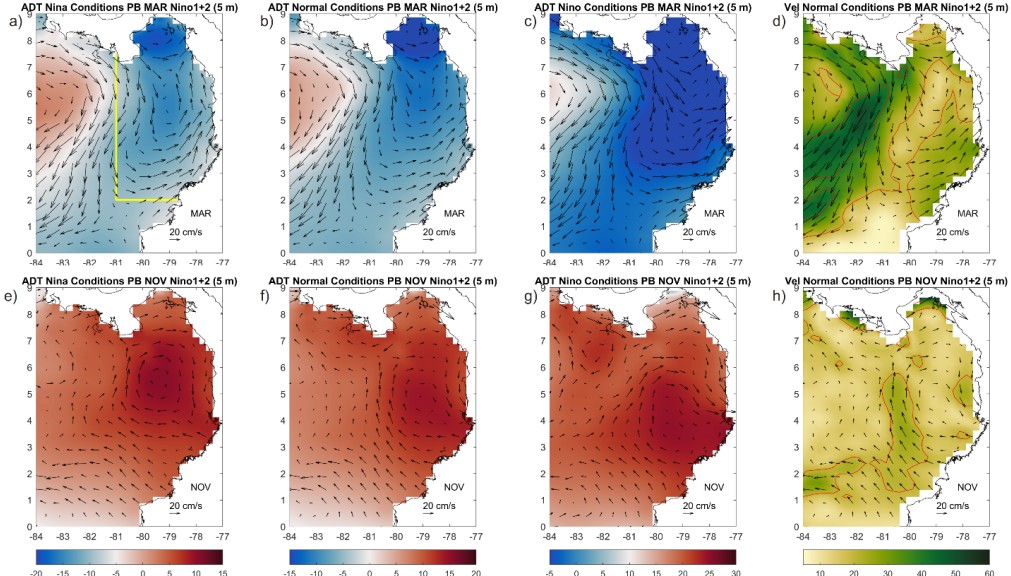

**Figure 3. Monthly averaged ADT anomalies (color scale in cm) and associated geostrophic circulation (vectors) for the Panama Basin. Seasonal changes are indicated within two months: March (a,b,c) and November (e,f,g). ENSO-related conditions (based on**
**Niño 1+2) shown are: normal (b,f), negative ENSO (a,e) and positive ENSO (c,g). Each condition has a different color range to highlight the ADT regional gradients, responsible for the geostrophic circulation. ADT Anomalies are computed subtracting the ETP regional mean during the 1993-2019 period (66.0 cm). Geostrophic vectors and their speed (color scale in ms$^{-1}$) for normal conditions are included for the same months (d,h) with 20 and 40 cms$^{-1}$ contours included. In panel (a) the yellow line indicates the Panama Bight area used to compute regionally-averaged time series shown in Figure 5a to d.**

In this season, the Panama Bight is dominated by a cyclonic circulation (Figure 3b), in response to the ADT drop (Figure 5a

black line), forced by the Panama wind jet (which produces Ekman transport to the west), and the corresponding upwelling

intensification (SST decrease and SSS increase as seen in Figure 4b and h). Besides, SSS increase in the Bight due to

precipitation and river outflow reduction during this season, contributing to the ADT drop (Figure 5). The relatively stronger

cyclonic circulation is composed by the Panama Jet Surface Current (~81° W) with speeds >40 cm s$^{-1}$ (Figure 3d), which

extends from the Azuero Peninsula to the southwest, turning into a westward flow at ~85°W (Figure 2b), merging at ~90° W

with the SEC. Besides, a limb of the Panama Jet Surface Current turns into an eastern flow ~2-3°N reaching the coast, where

it becomes the northerly Colombia Coastal Current, distinguishable from ~1-7° N. Interestingly, in the Panama Gulf the

cyclonic gyre does not close, as eastward currents are observed the entire year, regardless of the seasonal ADT variation.

Seasonal ADT, SST and SSS variations in the Panama Bight are strong and respond to local dynamics, which differ from

those from the ETP (Figure 5 and Figure A3 comparison). Besides, seasonality in the spatially averaged ETP time series



results from a combination of different ocean patterns observed in this region as the SEC, NECC, Costa Rica Dome, southern Cold Tongue and reverse circulation in the Panama Bight. Therefore, in Section 3.5 we explore ETP circulation patterns and their seasonality using SOM analysis.

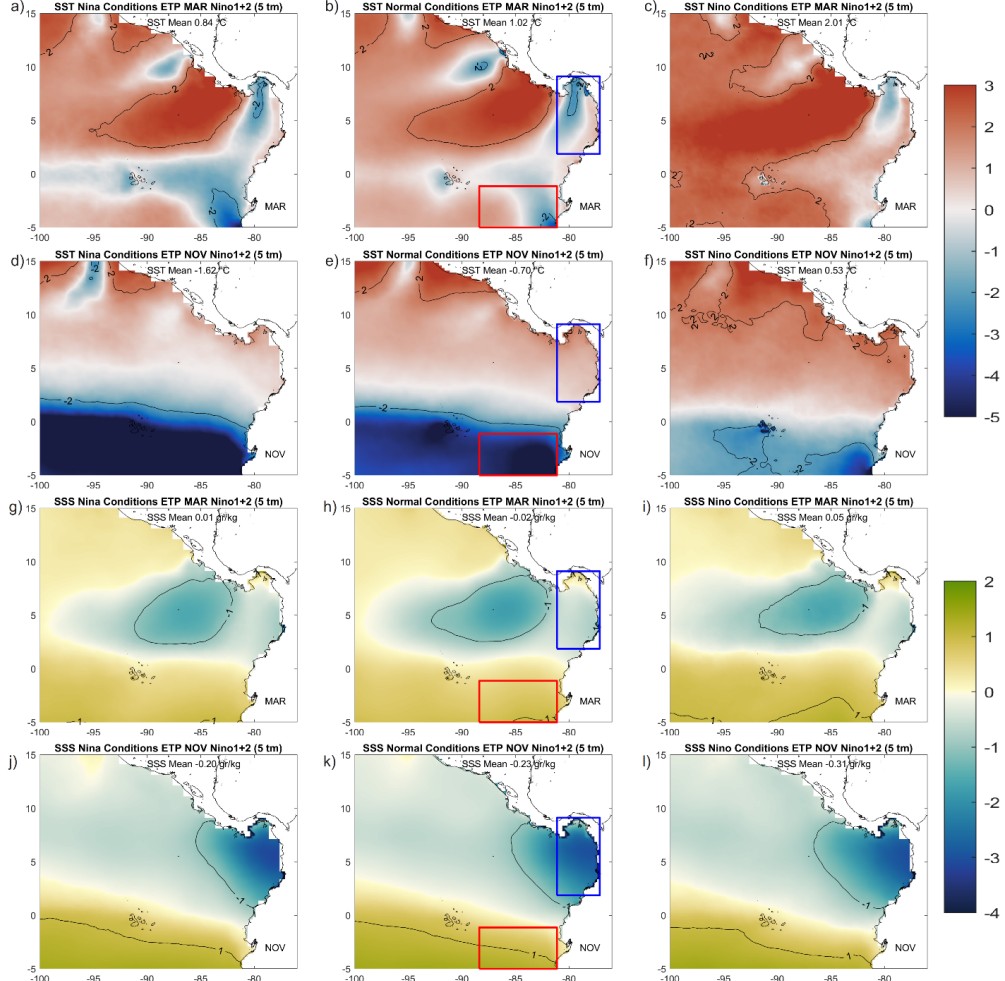

**Figure 4. Monthly averaged anomalies of Sea Surface Temperature (SST – a to f) in °C and Sea Surface Salinity (SSS - g to l) in gr kg⁻¹ for the Eastern Tropical Pacific – ETP. Seasonal changes are indicated within two months: March (a,b,c and g,h,i) and November (d,e,f and j,k,l). ENSO-related conditions (based on Niño 1+2) shown are: negative ENSO (first column), normal (second column) and positive ENSO (third column). Anomalies are computed subtracting the ETP regional mean during the 1993-2019 period (26.6 °C and 33.8 gr kg⁻¹). The regional average of the anomalies for each month is shown in the upper right of each panel, which coincides with time series values in Figure A3 c,d.**



We also compared geostrophic currents seasonality from ADT with the annual MDT and mean temporal SOM circulation (Figure A1). West of ~84°W, ADT and geostrophic circulation from July to December coincides with the annual MDT and mean SOM circulation. In contrast, the ADT and circulation in the Panama Bight does not coincide with the annual MDT or SOM circulation in any of the two seasons. A remarkable difference is observed in the strong MDT cyclonic currents seen in

the Panama Gulf, as this circulation is not observed at any month of the year in ADT. This eastward flow is shown in the mean temporal SOM.

### 3.3 Variations in the Panama Bight seasonal circulation related to ENSO

In this section we assess the influence of ENSO in the ETP and Panama Bight seasonal ocean dynamics. Only 2 years corresponding to the strong positive ENSO condition of 1997-1998 and 2015-2016 present this phase during January and

February (Figure 5f). Although this is a small number of outputs, we believe that even for these months, results accurately indicate El Niño conditions since there are two dominant seasonal dynamics in all the variables assessed, which are observed in the similar patterns from January to April or from May to December. For example, in the first quarter of the year, the 12 available positive ENSO months (Table 1), show very similar ocean dynamics during this season.

In the ETP, El Niño events increase SST, while the negative phase reduces it; in the contrary, a small interannual variation is

observed in SSS (Figure 4 and Figure A3). As a consequence, changes in temperature dominate a region-wide sea level rise during El Niño and sea level drop during La Niña in the entire ETP, including the Panama Bight (Figure 5). We assess if such sea level changes affect the seasonal circulation patterns observed in normal conditions (Section 3.2). At this point we want to remark that surface currents respond to ADT gradients and not to basin-wide sea level variations. Therefore, we highlight ADT gradients in all the ENSO-related conditions shown in Figure 2 and Figure 3, by shifting the ADT colour

limits but maintaining a 35 cm range. For negative/positive ENSO, ADT colour limits are shifted -5 cm/+10 cm with respect to the normal months.

Both El Niño and La Niña seasonal circulation patterns in the ETP are very similar to the circulation observed in normal months. From June to November, the relatively higher ADT anomalies in the 1° to 7° N band, extending to the Colombian coast persist (Figure 2). Besides, the anticyclonic circulation in the Panama Bight is also observed (Figure 3). Similarly, from

January to April, the Papagayo and Panama wind jets forcing is observed in all ENSO conditions, including the Costa Rica dome, the cyclonic circulation in the Panama Bight and the anticyclonic circulation between them. Therefore, a seasonal distinctive and opposite circulation pattern dominates the Panama Bight regardless of the ENSO-related ADT mean shifts. Besides, large interannual differences in the Panama Bight circulation speed are not observed (Figure 5b).

We also assess if the small interannual variations observed in the seasonal circulation, stand when other ENSO indexes are

used. For this purpose, we use the Niño 3 index to determine positive and negative ENSO anomalies. Although positive/negative ENSO months show large differences between the two indices (comparison of Figure 5f and Figure A3f), geostrophic currents seasonality does not change, showing small interannual variations in both cases. This indicates that our results stand even if ENSO variability is assessed from an open-ocean region.





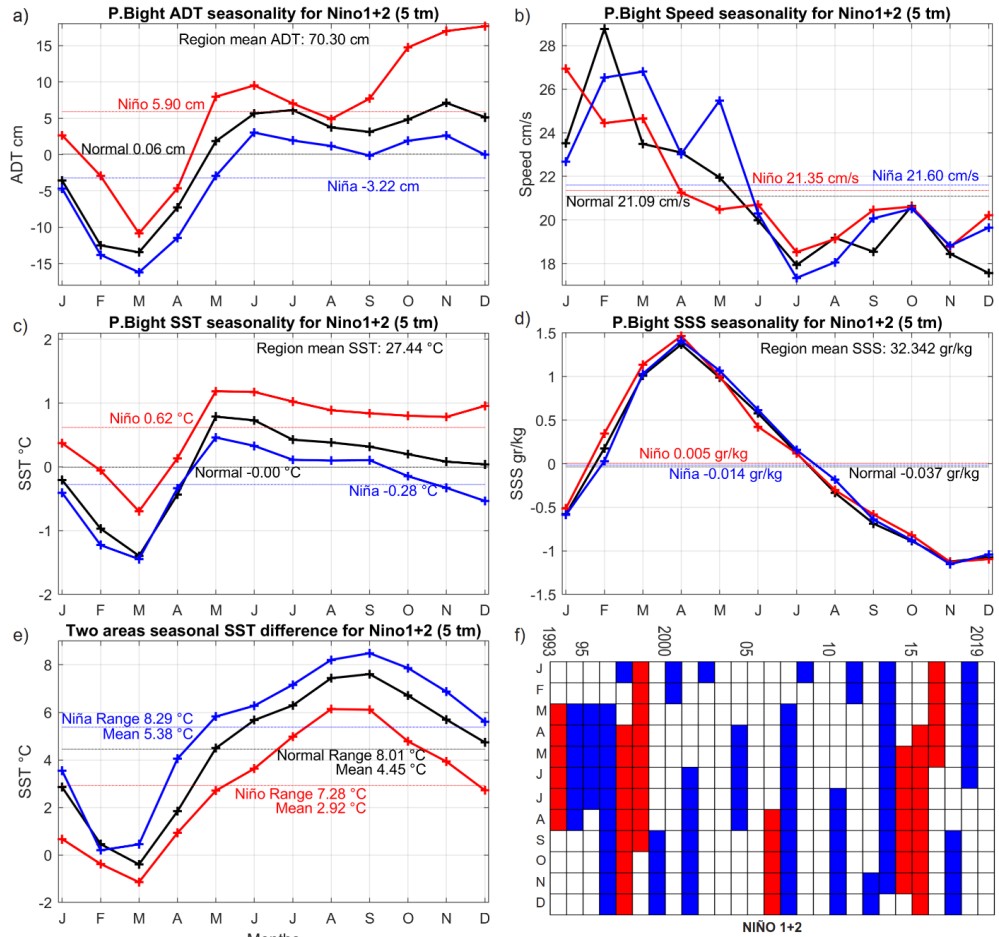

**Figure 5. Monthly spatially averaged values in the Panama Bight for 1993-2019, differentiating three ENSO-related conditions (based on Niño 1+2): normal (black), negative (blue) and positive (red), showing each time series annual mean. a) Absolute Dynamic Topography (ADT) anomalies in cm. b) Geostrophic currents speed in cm s⁻¹. c) Sea Surface Temperature (SST) anomalies in °C. d) Sea Surface Salinity (SSS) anomalies in gr kg⁻¹. Anomalies are computed subtracting the corresponding regional and temporal mean (value shown in the top of each panel). e) Monthly SST difference between the warm Panama Bight and the Cold Tongue areas shown in Figure 4. f) Matrix indicating the monthly ENSO-related condition between 1993-2019, based on Niño 1+2 region (Section 2). Normal months in white, negative and positive ENSO conditions in blue and red respectively.**

### 3.4 Regionally averaged seasonal and interannual variations in the Panama Bight

The assessment of seasonal and interannual regionally averaged ADT variations is important to evaluate sea level extremes that severely affect the coastal zone. Results in the previous section outlined that ENSO does not significantly affect ADT



gradients in the study area and thus does not affect the dominant seasonal circulation in the ETP and Panama Bight. However, ENSO affects the ADT regional mean, which we study in more detail here. As previously mentioned, the ETP 27-year regional ADT mean is 66.0 cm above the geoid, while for the Panama Bight this value is 70.3 cm. ETP, spatially averaged time series (Figure A3), responds to the combination of different circulation patterns (Figure 2). Mean ADT interannual variations are crucial to understand ENSO effects in regional sea level. The higher ADT mean corresponds to

the El Niño conditions (8.9 cm), while the lower is for La Niña (-3.8 cm). This difference is clearly related to the warmer SST during positive ENSO and relatively colder SST during negative ENSO conditions. Small interannual changes are observed in SSS, which do not significantly modify the seasonal signal. Besides, in the ETP, differences in geostrophic velocities are not noticeable during the three ENSO-related conditions. The annual mean differences in the three conditions are below 0.3 cm s$^{-1}$, (all of them around 22.9 cm s$^{-1}$). These results support the small impact in circulation (mean speed) due

to ENSO variability in the ETP.

The regionally averaged ADT time series in the Panama Bight responds to the reverse oceanic circulation pattern observed through the year (Figure 3), whose seasonality is not strongly affected by ENSO (Figure 5). Lowest ADT occurs from January to April, indicating the sub-regional sea level drop due to the cyclonic circulation forced by the Panama wind Jet. The upwelling intensification during these months is clearly observed in the SST drop and SSS increase. On the contrary,

from June to December ADT do warmer and fresher surface waters characterize higher and less variable, indicating the anticyclonic circulation in the sub-region. ADT seasonal range is 20.6 cm in the Panama Bight during normal months, which coincides with the seasonal cycle previously reported from tide-gauges and form a regression fitted to altimetry data (Dimar, 2020 - Chapter 5). Large seasonal differences are also observed in the geostrophic currents velocity. Stronger currents occur from January to April, while slower and less variable currents from June to December. Therefore, the cyclonic circulation

associated with the Panama wind jet, is also the fastest. Seasonal speed range in normal conditions is 11.2 cm s$^{-1}$, between February and December, thus about half the mean speed (Figure 5b).

ADT seasonality is similar for all ENSO-related conditions; therefore seasonality dominates over the interannual sea level shifts in the Panama Bight (Figure 5a). In this region, the ADT annual mean is higher (5.9 cm) during the El Niño condition, while lower during La Niña (-3.2 cm), which is a smaller difference than what was observed for the ETP. Note that the ADT

annual mean during El Niño is ~3 cm lower in the Panama Bight than in the ETP due to a weaker ocean warming in the Bight (0.56 °C below the ETP annual mean). Largest seasonal ADT range is found during El Niño in the Panama Bight (28.5 cm), as a consequence of a larger temperature increase from October to December due to El Niño peak at the end of the year. Besides, currents mean speed and seasonality in the Panama Bight are very similar in the three ENSO-related conditions (Figure 5b) indicating that ENSO phenomena affect mainly the sub-regional sea level and not the circulation patterns (mean

speed).

SSS has a strong seasonal cycle (up to 2.59 gr kg$^{-1}$ during El Niño) with small interannual variations in the Panama Bight (Figure 5d). This was unexpected as literature indicates that ENSO affects locally precipitation and river runoff, which will also affect SSS. To explore the reasons behind the small interannual SSS variations, we assess the SST spatial gradient





between the southern Cold Tongue and the Panama Bight (areas shown in Figure 4), as this gradient modulates the Choco

surface wind jet.

SST monthly means in the Cold Tongue show larger seasonal variability than SST in the Panama Bight (Figure A3e), with warmer months from January to April (ITCZ at its southernmost position produce weaker upwelling in the Cold Tongue). In normal ENSO conditions, only in March the SST in the Panama Bight is colder than the SST in the Cold Tongue. In both areas, El Niño (La Niña) conditions show warmer (colder) SST, however, shifts from normal conditions are larger in the

Cold Tongue. Note that coldest SST in the Cold Tongue occurs in September under La Niña conditions, which would indicate stronger upwelling as a consequence of stronger southern Trade winds.

From May to December the SST differences between the Panama Bight and Cold Tongue is larger (Figure 5e), which coincides with the intensification of the Choco surface wind jet in the former area. These results are in accordance with literature (Section 2). Besides, a larger SST difference between the Panama Bight and the Cold Tongue occurs during La

Niña, with an annual mean of 5.4°C and seasonal range of 8.3°C; while they are smaller in El Niño conditions (2.9°C and 7.3°C). These results indicate that in positive ENSO conditions, the Choco surface wind jet will be weaker than in normal conditions (smaller SST differences between the two areas from May to December), while the opposite will happen in negative ENSO conditions.

During El Niño conditions, warmer SST enhances precipitation in the Panama Bight which is expected to reduce SSS.

However, we speculate that small SSS ENSO-related variations found in the Panama Bight are due to compensating mechanisms acting differently in the two seasons. In the first quarter of the year, positive ENSO strength the Panama surface wind jet (e.g. Sayol et al., 2022), enhancing upwelling in the Panama Bight, what would increase SSS, which would be reduced (compensated) due to enhanced precipitation. From May to December, a weaker Choco surface wind jet reduces the moisture inshore transport to the Panama Bight, therefore compensating precipitation increase due to warmer SST. The

opposite mechanism will occur in La Niña conditions. Bear in mind that small interannual SSS variations in the Panama Bight, do not necessarily indicate small ENSO-related variations in coastal precipitation, what is an active topic of study (e.g. Sayol et al., 2022).

We also assess results from this section using the Niño 3 index (not shown). We found a larger difference among the ADT annual means for the three ENSO-related conditions in the Panama Bight. The mean for El Niño is 6.5 cm and for La Niña -

3.5 cm. However, regional ADT and current speed seasonality is very similar to the results using the Niño 1+2 index. Therefore, we did not find large differences in the results we report in this section, when ENSO variability is assessed from a larger equatorial area in the open ocean. However, we believe that using the Niño 1+2 index is more appropriate to assess circulation variations in the Panama Bight, as dynamics in this region (e.g. Cold Tongue) affect our study area (e.g. Choco wind jet).



**3.5 Circulation patterns in the Eastern Tropical Pacific from SOM Analysis**

Results from the SOM analysis in the temporal domain support previous findings in the ETP which indicate that surface currents' strong seasonality dominates over interannual variations associated with ENSO; in the contrary, ADT is strongly modulated by El Niño (Figure 6a). The seasonal cycle explained variance in the zonal and meridional currents is 40% and 48% respectively when averaged the six neurons, while their correlation mean with Niño 3 is 0.38 and 0.26 respectively (Table 2). Conversely, the seasonal cycle mean explained variance in ADT is only 10%, while the mean correlation with Niño 3 is 0.87. Slightly weaker mean correlations are found with Niño 3.4 (0.84) and Niño 1+2 (0.78).

Zonal currents dominate the circulation in the ETP with a mean value one order of magnitude larger than meridional currents (Table 2). A comparison between the temporal SOM spatial distribution (Figure 6d) and main currents in the ETP (Figure A1) allow us to identify the second neuron with the SEC and the fifth neuron with the NECC, extending to the east as part of the southern limit of the Costa Rica dome. The former has the strongest eastwards flow (-34.0 cm s$^{-1}$) and the latter the strongest westwards flow (18.8 cm s$^{-1}$). Besides, neuron six seems to be related to the westward circulation in the Costa Rica Dome (northern side) and Cold Tongue (west of 89°W), characterized by small mean ADT. Note that the strongest ADT (U) correlation with Niño 3 is 0.92 (~0.46) in neurons 2 (SEC) and 4 (SEC boundary). Besides, neurons 2, 4 and 6, which are mainly westward, can reverse and become eastward during strong El Niño events, such as the one in 1997-98 (Figure 6b). On the contrary, there is no significant correlation between U and Niño 3 in neuron 5 (NECC). Therefore ENSO effects on the ETP currents are not homogeneous.

**Table 2. Temporal SOM 3x2 neurons (Neu) of variability for Absolute Dynamic Topography (ADT), Zonal (U) and Meridional (V) currents. Mean, percentage of explained variance of the seasonal cycle (%EV SC) and significant correlations with Niño 3 are shown. Number of months of occurrence of the different ENSO conditions (Niño 3) for the six spatial SOM neurons are included.**

| | ADT | | | Zonal Current | | | Meridional Current | | | No. Months | | |
|---|---|---|---|---|---|---|---|---|---|---|---|---|
| | Mean (cm) | %EV SC | Corr | Mean (cm s$^{-1}$) | %EV SC | Corr | Mean (cm s$^{-1}$) | %EV SC | Corr | Norm | Niño | Niña |
| Neu1 | 77 | 19 | 0.78 | 0.3 | 36 | 0.29 | -0.9 | 38 | -- | 36 | 4 | 21 |
| Neu2 | 70 | 3 | 0.92 | -34.0 | 28 | 0.47 | -2.2 | 37 | -- | 34 | 6 | 6 |
| Neu3 | 72 | 17 | 0.85 | 9.6 | 58 | 0.23 | -0.4 | 28 | 0.16 | 28 | 1 | 13 |
| Neu4 | 65 | 2 | 0.92 | -24.8 | 21 | 0.46 | -1.7 | 77 | 0.22 | 24 | 24 | 1 |
| Neu5 | 67 | 14 | 0.87 | 18.8 | 79 | -- | 0.2 | 34 | 0.29 | 36 | 0 | 41 |
| Neu6 | 60 | 2 | 0.89 | -15.5 | 18 | 0.43 | -1.1 | 74 | 0.36 | 37 | 12 | 0 |
| Mean /Total | | 10 | 0.87 | | 40 | 0.38 | | 48 | 0.26 | 195 | 47 | 82 |



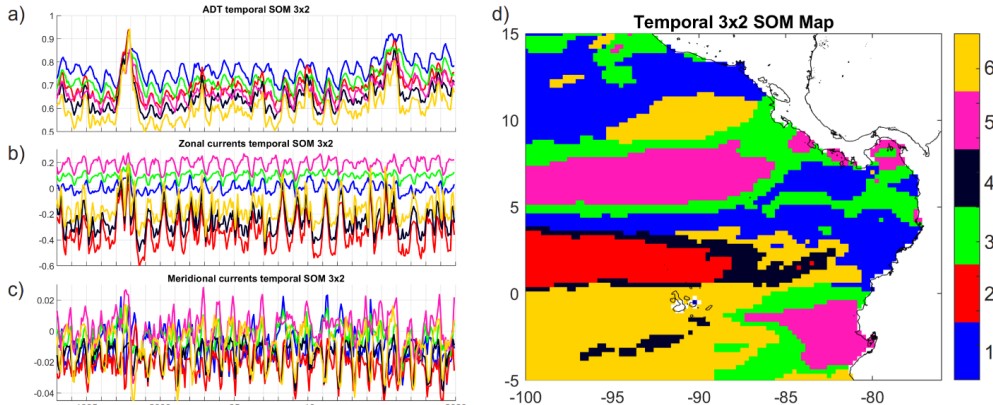

**Figure 6. a) Absolute Dynamic Topography (m), b) zonal and c) meridional geostrophic currents (m s⁻¹) time series from 1993 to 2019 which results from a 3x2 SOM temporal analysis in the Eastern Tropical Pacific. d) Sub-regions represented by the six temporal SOM patterns.**


Neurons 1 and 3 dominate the Panama Bight (Figure 6d). Neuron 1 is the only one in which mean (southwards) meridional currents are larger than mean zonal currents. However, both neurons have large seasonality, therefore they regularly change direction (except U in neuron 3), which agrees with the reverse seasonal circulation forced by the surface wind jet in the Panama Bight (Figure 3).

The six SOM neurons in the spatial domain, indicate mainly two different circulation patterns (Figure 7a). Neurons 1 and 2, show the dominant circulation from January to May (Section 3.2), where the effect of the Panama and Papagayo surface wind jets is observed in a stronger Panama Jet Surface Current and northern boundary of the Costa Rica dome. On the contrary, the other neurons show the dominant circulation from June to December. However, neurons 3 and 4 differ from 5 and 6 mainly in the Panama Jet Surface Current, as in the former neurons it extends towards the south from the Azuero

Peninsula, while in the latter neurons, this jet disappears due to the anticyclonic circulation that dominates the Panama Bight. Therefore, neurons 3 and 4 seem to be transitional circulation patterns. The temporal occurrence, shown in the BMU (Figure 7b), corroborates the former description. Neurons 1 and 2 always occur between January and May (79.3% of the months), while neurons 5 and 6 always occur between June and December (66.7% of the months). Neurons 3 and 4 together are observed in 91 months, 71.4% occurring in December, January, May and June, during the two seasons' transition.

Regional mean ADT shows large differences between the odd and even neurons. Odd neurons show smaller ADT spatial mean (63.5 to 63.8 cm) than even neurons (73.3 to 73.6 cm). This ADT shift corresponds to what is expected from La Niña and El Niño conditions respectively (Figure A3). Furthermore, odd and even neurons do not show large differences in circulation (when compared in the same row in Figure 7a), except for the strongest westward currents in the Cold Tongue and SEC, as expected from La Niña conditions. These results corroborate that in the ETP ENSO strongly affects the mean

ADT, but not the circulation seasonality (Section 3.3). To highlight this relation, we combine the BMUs with the ENOS





prevailing condition (Figure 7b). From 82 months with La Niña condition, 91.5% coincide with neurons 1, 3 and 5. From 47 months with El Niño condition, 89.4% coincide with neurons 2, 4 and 6 (Table 2). Similar results were obtained when ENOS variability is calculated with Niño 1+2, however the relation with the spatial SOM neurons deteriorate, as neurons 1, 3 and 5 coincide with 83.1% of La Niña months, while neurons 2, 4 and 6 coincide with 76% of El Niño months.

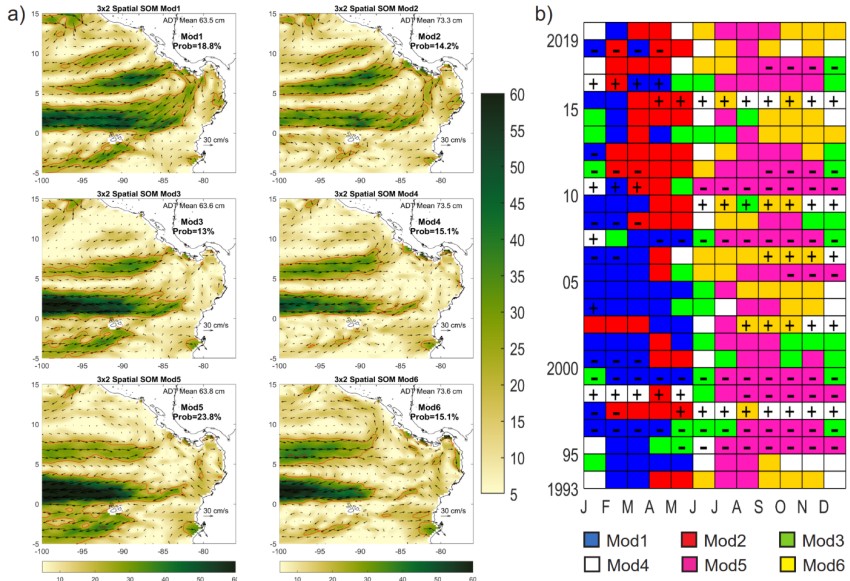


**Figure 7. a) Spatial patterns of geostrophic speed (color scale) and currents (vectors in cm s$^{-1}$), which results from a 3x2 SOM spatial analysis in the Eastern Tropical Pacific, computed from 1993 to 2019. 20 and 40 cm s$^{-1}$ contours are included. b) Monthly Best Matching Unit (BMU) of SOM patterns, indicating negative (-) and positive (+) ENSO-related conditions, based on Niño 3 region.**

## 4 Summary and final remarks


A reverse seasonal ocean circulation in the Panama Bight has been assessed using 27 years of ADT in accordance with previous works that reported a similar feature using SLA and hydrographic data (Devis-Morales et al., 2008; Rodríguez Rubio et al., 2003).

The mean circulation in the ETP (east of 100°W) was analyzed from the MDT altimetry product (1993-2012 period) and a

Self Organizing Map analysis. Small differences are observed west of ~82°W. In the Panama Bight, MDT shows the cyclonic circulation when the Panama surface wind jet dominates the region (Figure A1). However, the strong counterclockwise circulation in the Panama Gulf (northern part of the Bight) does not correspond to any of the observed monthly ADT circulation patterns (Figure A2). On the contrary, the mean temporal SOM circulation in the Panama Bight seems to better represent the eastward flow in the Panama Gulf showing the anticyclonic circulation pattern when the Choco





surface wind jet prevails. Therefore, the mean annual circulation is not adequate to represent the dominant seasonal circulation in the Panama Bight.

The seasonal circulation assessment in the Panama Bight shows that from January to April, a stronger cyclonic circulation responds to the ADT drop (Figure 3), produced by upwelling forced by the Panama surface wind jet, which also reduces SST and increases SSS (Figure 5). From June to December, a weaker anticyclonic circulation responds to the ADT rise (density

reduction), produced by convergence of surface warm waters dragged by the Choco surface wind jet, which also transport moisture inshore, enhancing precipitation and river outflow, which reduces SSS. Therefore, the Panama Bight has a strong seasonal variation in SST, SSS, sea level and circulation. The variations of these physical factors affect the biosphere, as chlorophyll-a availability in the basin is modulated by these changes (Corredor-Acosta et al., 2020). In the ETP seasonal circulation variations do not change as much as in the Panama Bight (Figure 2), showing the zonal presence of the SEC and

NECC west of ~90° W, as permanent features; although the cyclonic Costa Rica Dome is also a permanent feature, its intensity varies seasonally (Figure 7).

We further assess ENSO effect on seasonal circulation and sea level in the study area using the coastal Niño 1+2 index. However, if the open ocean Niño 3 index replaces this index, results are very similar. We find that the seasonal SST, SSS, sea level and circulation patterns are not largely modified by ENSO positive or negative phases (Figure 5). On the contrary,

the ETP and Panama Bight regionally averaged SST and sea level are shifted by ENSO, with a mean SST and ADT increase in the positive phase and a mean SST and ADT decrease in the negative phase. The mean ADT (SST) difference between the two ENSO conditions is 12.8 cm (1.8 °C) in the ETP, and 9.1 cm (0.9 °C) in the Panama Bight. Still, seasonal ranges of these two variables in the Panama Bight (up to 28.5 cm during El Niño and 2.2°C during regular conditions) dominate over the ADT and SST interannual shifts.

In the western-central tropical Pacific, La Niña (El Niño) reduces (increases) freshwater flux, affecting SSS interannual variability (Zhang et al., 2012). On the contrary, SSS in the ETP is not affected by ENSO, maintaining its strong seasonal cycle in the Panama Bight (range up to 2.6 gr kg$^{-1}$ during El Niño). To understand this result, we assess the SST gradient between the warm Panama Bight and the Cold Tongue, as it modulates the Choco surface wind jet (Poveda and Mesa, 2000). We speculate that this small ENSO effect on SSS is due to a local compensating mechanism in the Panama Bight (Section

3.4). ENSO increases the SST, which enhances precipitation (reduces SSS) the entire year. This effect is compensated differently in the two observed seasons. In the first quarter of the year, ENSO affects the Panama wind jet strength, which enhances upwelling and increases SSS. In the rest of the year, positive ENSO reduces the Choco wind jet strength, decreasing inshore moisture transport and precipitation (increasing SSS). Such complex mechanisms should be studied in more detail as well as its influence in coastal precipitation and river runoff.

We use six spatial and temporal neurons obtained from SOM analysis to assess the ETP circulation, as well as the ADT seasonal variability and its relation to ENOS conditions. SOM results confirm the opposite seasonal circulation patterns in the Panama Bight, which are not strongly affected by ENSO. Results also support the strong ENSO effect in the ETP mean



ADT, increasing (decreasing) mean sea level during El Niño (La Niña). Besides, ENSO seems to be strongly related to the SEC while it does not show a significant relation with the NECC.

The seasonal description of the reverse circulation in the Panama Bight, as well as the ENSO-related interannual variations is useful to assess regional fluctuations in ocean dynamics. These dynamic changes will have implications in maritime activities such as navigation, but also, this seasonality and interannual variations might affect local climate through ocean-atmosphere fluxes, determine biosphere cycles, force extreme sea levels and enhance erosion, affecting coastal communities. For example, sea level extremes, that affect the coastal areas in the Panama and Colombia coasts in the Pacific Ocean, will

increase their flooding probability in December under El Niño conditions, when the monthly mean sea level is 17.7 cm higher than the multiannual ADT mean. Therefore, the study of these air-sea interactions, their temporal variations and relation with global warming in the region should be encouraged.

### Acknowledgments

A. Orfila thanks financial support from Projects LAMARCA (PID2021-123352OB-C31) funded by MICIN/AEI
/10.13039/501100011033/ FEDER, UE and Tech2Coast (TED2021-130949B-I00) funded by MCIN/AEI/ 10.13039/501100011033 and by EU "NextGenerationEU"/PRTR». I. Hernández-Carrasco is supported by the TRITOP Project (UIB2021-PD06) funded by the Universidad de las Islas Baleares – FEDER (UE)

### Author contribution

RR and AC conceived the idea of the study with the support of EG and CM. IH-C, AO and AC performed SOM assessment.
All authors contributed to data processing, figures preparation and analysis of results. RT prepared the manuscript with contributions from AO.

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




**Appendix A**

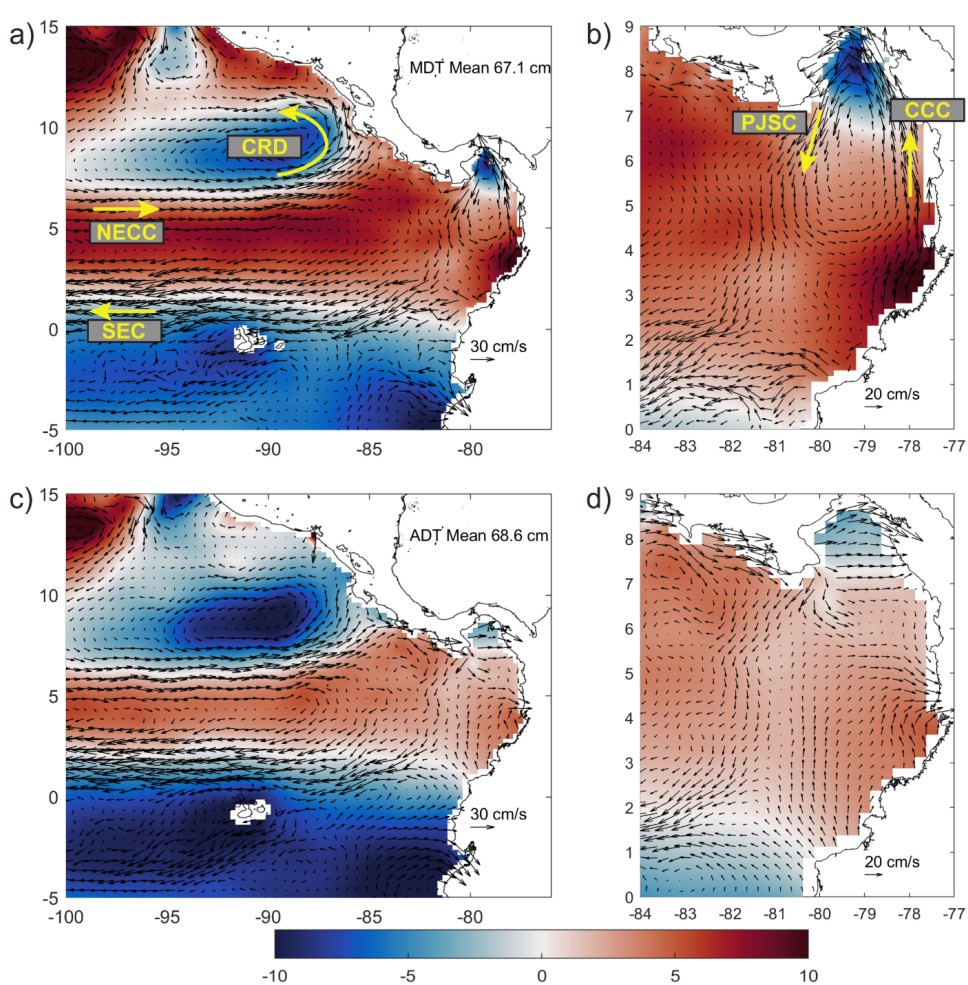

**Figure A1. Mean Dynamic Topography (MDT) anomalies (color scale) and associated geostrophic circulation (vectors) representative of the 1993-2012 period for the a) Eastern Tropical Pacific – ETP east of 100° W and b) the Panama Basin. MDT anomalies are computed subtracting the ETP regional mean (67.1 cm). SOM mean circulation showing Absolute Dynamic**
**Topography (ADT) anomalies (color scale) and associated geostrophic circulation (vectors) for c) Eastern Tropical Pacific – ETP east of 100° W and d) the Panama Basin. SOM anomalies are computed subtracting the ETP regional mean (68.6 cm). South Equatorial Current (SEC), North Equatorial Counter Current (NECC), Costa Rica Dome (CRD), Colombia Coastal Current (CCC) and the Panama Jet Surface Current (PJSC) are shown.**



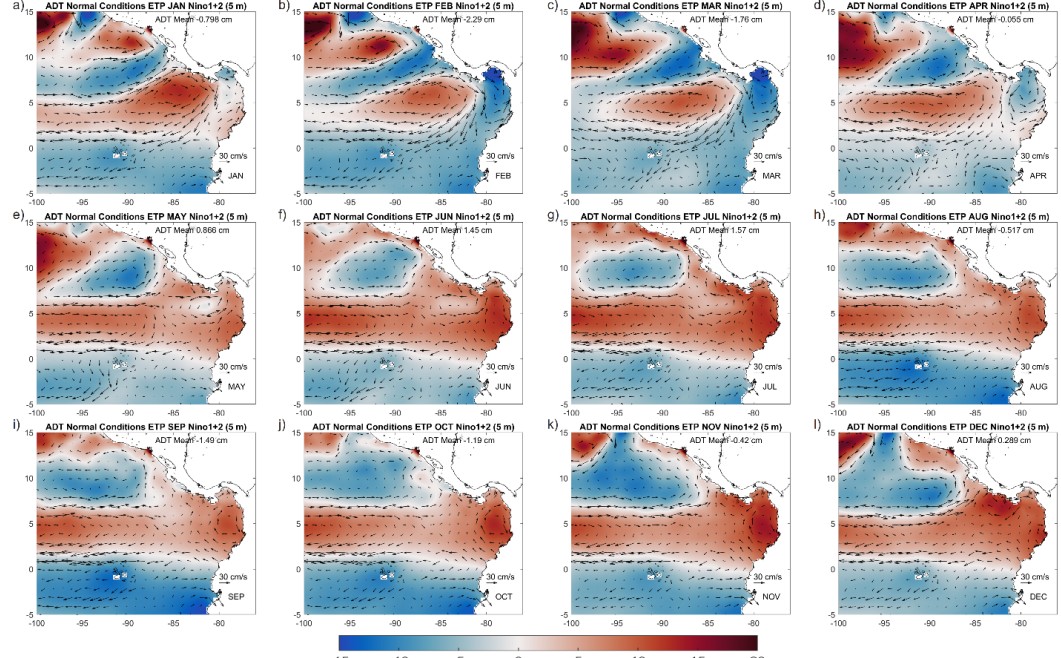


**Figure A2. Monthly averaged ADT anomalies (color scale in cm) and associated geostrophic circulation (vectors) for the Eastern Tropical Pacific – ETP. ENSO-related normal conditions (based on Niño 1+2). ADT Anomalies are computed subtracting the ETP regional mean during the 1993-2019 period (66.0 cm). The regional average of the ADT anomalies for each month is shown in the upper right of each panel, which coincides with the black line in Figure A3a time series.**




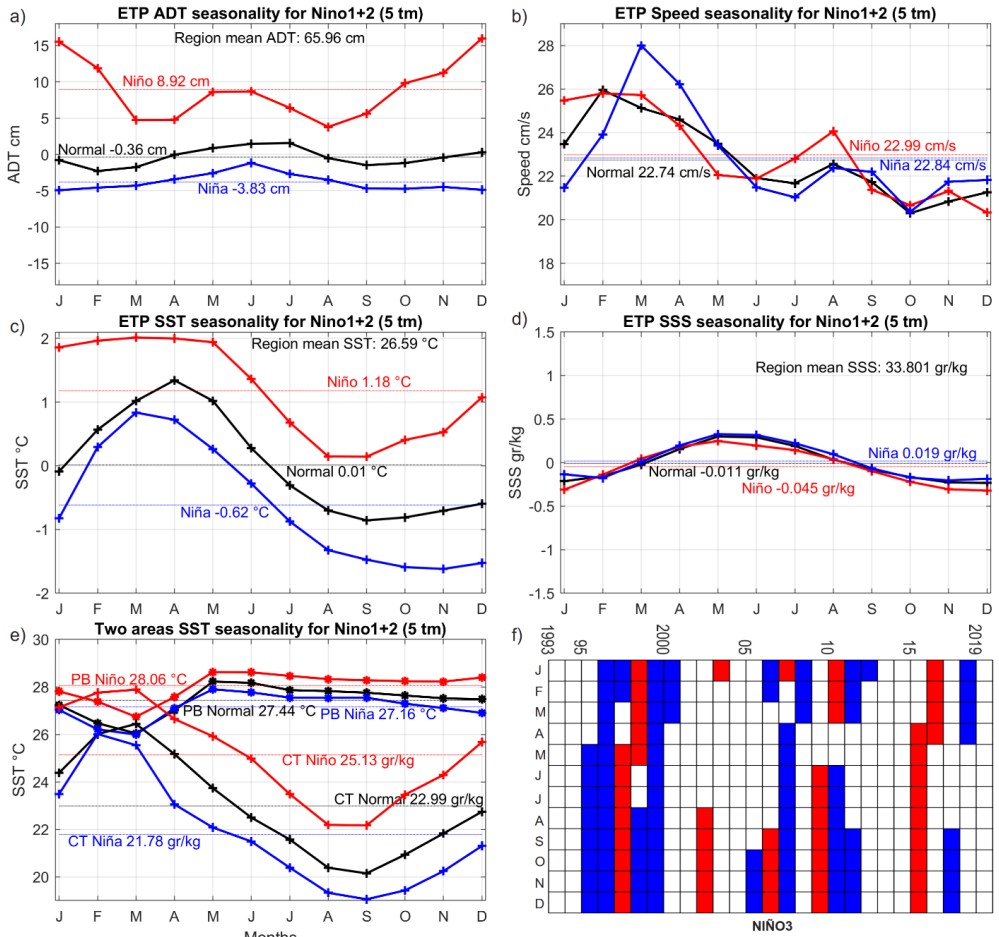

**Figure A3. Monthly spatially averaged values in the Eastern Tropical Pacific for 1993-2019, differentiating three ENSO-related conditions (based on Niño 1+2): normal, negative and positive, showing each time series annual mean. a) Absolute Dynamic Topography (ADT) anomalies in cm. b) Geostrophic currents speed in cm s⁻¹. c) Sea Surface Temperature (SST) anomalies in °C.**
**d) Sea Surface Salinity (SSS) anomalies in gr kg⁻¹. Anomalies are computed subtracting the corresponding regional and temporal mean (value shown in the top of each panel). e) Monthly SST of the warm Panama Bight (PB) and the Cold Tongue (CT) areas shown in Figure 4. f) Matrix indicating the monthly ENSO-related condition between 1993-2019, based on Niño 3 region (Section 3). Normal months in white, negative ENSO in blue and positive ENSO in red.**