# Peer review of "Seasonal and ENSO-related ocean variability in the Panama Bight"

_EGUsphere, 2022_

## Author Comment (AC1)

**"Seasonal and ENSO-related ocean variability in the Panama Bight" by Rafael R. Torres, Estefanía Giraldo, Cristian Muñoz, Ana Caicedo, Ismael Hernández-Carrasco & Alejandro Orfila.**

We would like to thank the reviewer for their helpful and constructive comments to improve our paper. We have tried to respond to all of the comments and we hope that the paper has improved so as to be accepted for publication.

**Referee #1:**
*General Scientific Comments:*
This paper addresses some unresolved issues related to the mean, seasonal and interannual variability in the circulation, SST and SSS in the Panama Bight, also extending the analysis to the larger Eastern Tropical Pacific (ETP) for context. As such, it builds on previously published results, with new results concerning the degree of variability in the circulation caused by ENSO changes (minor). It confirms some previous results, such as the reversing seasonal states of cyclonic (boreal winter) and anticyclonic circulation (June-December); seasonal and interannual changes in SST but only seasonal changes in SSS; seasonal and interannual changes in the Costa Rica Dome and NECC. It finds that during the positive and negative conditions of the ENSO cycle (El Niño and La Niña periods), the circulation in the Panama Bight is not greatly changed. Temperature and sea level means are affected, but the circulation patterns remain largely the same. I believe this is a new finding and may surprise some people.

*Specific Scientific Comments, Related to the "Principal Criteria"*
**Scientific Significance: "Good" (2)**. The paper mostly confirms or clarifies aspects of the circulation that have been discussed by others. This is done using 27 years of altimeter data, a much more comprehensive data set than has been used previously, making this a worthwhile but not astounding contribution. However, two aspects of the Panama Bight circulation that were previously uncertain are not really resolved.

(1) One feature is the flow along the Colombia Coast, which is normally reported as forming the poleward Colombia Current (strongest in boreal winter). Even the more recent references do not show this feature clearly and sometimes lament the lack of data. On Line 75, the authors quote Kessler (2006) as saying that there is no consensus about this phase of the circulation (the anticyclonic phase, which should have an equatorward flow). During the period of anticyclonic flow documented in the present paper, flow along the Colombia coast should be equatorward, rather than poleward. Indeed, from their pictures of the seasonal circulation in Figure 3 and A2, they should conclude that the flow is poleward in boreal winter north of ~1°N and equatorward in boreal summer-Autumn north of ~4°N. But they don't really discuss or clarify this previously disputed flow.

We thank the reviewer for his suggestion about looking in more detail the differences in the Panama Gulf currents between the MDT (westward) and ADT under all the three ENSO conditions (eastward).  As we explain latter, we decided to assess the ETP ADT and geostrophic currents using the "2-Sat" product:

(https://data.marine.copernicus.eu/product/SEALEVEL_GLO_PHY_CLIMATE_L4_MY_008_0 57/description), instead of the "All-Sat" database:

(https://data.marine.copernicus.eu/product/SEALEVEL_GLO_PHY_L4_MY_008_047/descripti on), which was used in the first version of the paper. Using the "2-Sat" product gave us more coherent results in the Panama Bight circulation patterns, as presented in all the corresponding figures of the new version of the paper. Using the "2-Sat" product confirms the permanent westward currents in the Panama Gulf, as well as the permanent northward Colombia Coastal Current, north of ~4°N (new figure 4). These results are in agreement with what is shown in the MDT circulation (new figure 2), and a paper we included in this version from Chaigneau et al. (2006), where a similar circulation is shown based on 25 years of satellite-tracked drifters trajectories in the Panama Bight. We discuss in a clearer way these results in the first paragraph of the Summary and final remarks section of the new version of the paper.

(2) The other (more concerning) uncertain aspect is the flow at the northern boundary of the Panama Bight, which is reported in older publications as westward/poleward, completing the cyclonic circulation, but which should shift from westward during the boreal winter cyclonic circulation to eastward during the boreal summer-autumn anticyclonic circulation. In the text and Figure A1, they show a mean westward current at the top of the Panama Bight (PB) in the 1993-2012 Mean Dynamic Topography (MDT) field produced by CMEMS. But when they form the monthly mean seasonal cycle using the 1993-2019 Absolute Dynamic Topography (ADT) fields, the currents at the top of the PB are eastward during all months. They use the SOM methodology to produce a mean from the ADT data and it also shows eastward velocities at the top of the PB. They could have formed a mean of the ADT monthly means and gotten the same result. They seem to accept the result of eastward flow, perhaps because the SOM analysis is a statistical method and gives this result. What they don't discuss is the difference in the reference periods used in the two products (see below). This is my main complaint about their scientific analysis.

Our complete answer to this comment is in the next response. We removed the mean temporal SOM results included in the first version of the paper.

**Scientific Quality: Fair (3)**
They need to go into more detail about the formation of the MDT, the altimeter sea level anomalies (SLA) and the ADT fields formed from the SLA+MDT.

The ADT fields are formed from the sum of the MDT and SLA, which are formed by subtracting the 1993-2012 mean of the sea surface height fields from each gridded field (including those during 2013-2019, outside the reference period). In principal, the mean of the velocity anomalies derived from the SLA over the period 1993-2012 should be zero and the mean of the ADT velocities over the period 1993-2012 should be the same as those derived from the MDT. But the

mean of the ADT velocities over the 1993-2019 period will differ from those from the MDT if the mean of the velocities over the 2013-2019 period are different from those during 1993-2012. This appears to be the case and we can conclude that the mean velocities during the 2013-2019 period at the top of the PB are eastward. The interpretation of this result is made more complicated by the fact that the number of altimeters has increased over time, giving the ADT fields more spatial resolution during the latter years than during much of the 1993-2012 period of the MDT. The CMEMS program produces a "2-Sat" ADT product (using just 2 altimeters during the entire time series) for use in climate studies, to try to eliminate this bias in the number of satellites. I assume that the authors used the "All-Sat" product (using all available altimeters – most people do this to get the greatest spatial resolution).

If the authors want to compare the MDT to the mean from the ADT and to look at the monthly means in comparison to the MDT, they should form the monthly seasonal means from the ADT over the period 1993-2012. They might even do the same thing using the "2-Sat" data set to see if it agrees with the "All-Sat" results. This should clarify whether the mean flow at the top of the Panama Bight is eastward or westward.

We want to thank the reviewer for this comment and suggestions, which have helped us to improve the results shown in the paper. We assessed the two suggestions. First, the possible differences in the Panama Gulf ADT geostrophic currents when calculated from different periods. Second, the comparison of the "2-Sat" Vs. "All-Sat" ADT products.

Differences in the Panama Gulf ADT (All-Sat) geostrophic currents in different periods.

We selected a meridional segment placed at 79.125°W, between 7 to 8.8°N (7 nodes at 0.25° - left in Figure R1), and calculated some statistics using the monthly zonal velocity for different time periods: a) 1993-2012. b) 2013-2019. c) 1993-2019. Period (a) coincides with the period used to assess the MDT (CLS-18 – Mulet et al., (2021)) product used in the paper.

Figure R1.

[Figure]

[Figure]

For the three periods, mean ADT zonal velocity is eastward (+) at all nodes in the Panama Gulf (Table R1). Therefore, zonal velocities in all periods are coherent with results shown in Figure 3 from the first version of the paper. We also assessed the number and percentage of months with westward currents ("Neg month") in each of the three periods. The largest percentage of negative months is 37% in latitude 8.125°N for the 2013-2019 period. The month with most negative values (5) is April (4).The mean of the 31 eastward monthly velocities in this 84-months period is -9.1 cm/s. Still, the mean of all velocities in this period is 4.1 cm s$^{-1}$, the smallest eastward mean zonal velocity shown in Table R1.

Table R1

| | ADT at longitude -79.125 | | | | | | | | | | | | | | | | | | | | |
|---|---|---|---|---|---|---|---|---|---|---|---|---|---|---|---|---|---|---|---|---|---|
| | 1993-2012 (240 months) | | | | | | | 2013-2019 (84 months) | | | | | | | 1993-2019 (324 months) | | | | | | |
| | All U | Neg month | | | Mode | | | All U | Neg month | | | Mode | | | All U | Neg month | | | Mode | | |
| Latitude | mean | No. | % | mean | Month | No. | | mean | No. | % | mean | Month | No. | | mean | No. | % | mean | Month | No. | |
| 7.125 | 14.3 | 36 | 15.0 | -8.3 | 4 | 11 | | 17.3 | 9 | 10.7 | -8.9 | 4 | 4 | | 15.1 | 45 | 13.9 | -8.5 | 4 | 15 | |
| 7.375 | 16.6 | 39 | 16.3 | -6.7 | 4 | 11 | | 18.0 | 11 | 13.1 | -8.2 | 4 | 4 | | 16.9 | 50 | 15.4 | -7.0 | 4 | 15 | |
| 7.625 | 15.7 | 33 | 13.8 | -7.9 | 4 | 7 | | 14.7 | 14 | 16.7 | -7.5 | 4 | 4 | | 15.5 | 47 | 14.5 | -7.8 | 4 | 11 | |
| 7.875 | 11.0 | 54 | 22.5 | -9.4 | 8 | 9 | | 6.4 | 28 | 33.3 | -7.0 | 4 | 5 | | 9.9 | 82 | 25.3 | -8.6 | 9 | 13 | |
| 8.125 | 11.5 | 59 | 24.6 | -10.2 | 8 | 10 | | 4.1 | 31 | 36.9 | -9.1 | 4 | 5 | | 9.6 | 90 | 27.8 | -9.8 | 8 | 12 | |
| 8.375 | 25.2 | 27 | 11.3 | -12.1 | 2 | 8 | | 17.7 | 17 | 20.2 | -12.7 | 2 | 6 | | 23.3 | 44 | 13.6 | -12.3 | 2 | 14 | |
| 8.625 | 36.4 | 29 | 12.1 | -18.0 | 2 | 13 | | 29.8 | 16 | 19.0 | -19.4 | 2 | 7 | | 34.7 | 45 | 13.9 | -18.5 | 2 | 20 | |

We did a similar exercise using the MDT CLS18 product (1993-2012). As spatial resolution is 0.125°, we assessed the zonal velocity at 14 nodes in two longitudes, 79.1875 and 79.0624, at the sides of the nodes used from ADT (right in Figure R1). We find westward currents in all the nodes (Table R2), as shown in Figure Aux1 from the first version of the paper. Therefore, in the Panama Gulf, zonal currents are opposite in the 1993-2012 period when compared MDT and ADT from the "All-Sat" product.

Table R2

| Latitude | MDT - CLS18 | |
| --- | --- | --- |
| | Longitude | |
| | -79.1875 | -79.0625 |
| 7.0625 | -2.3 | -2.5 |
| 7.1875 | -3.0 | -2.8 |
| 7.3125 | -10.1 | -8.0 |
| 7.4375 | -8.6 | -8.5 |
| 7.5625 | -15.0 | -18.6 |
| 7.6875 | -15.9 | -22.2 |
| 7.8125 | -7.4 | -10.9 |
| 7.9375 | -5.2 | -5.5 |
| 8.0625 | -15.8 | -14.3 |
| 8.1875 | -21.8 | -28.0 |
| 8.3125 | -18.6 | -18.9 |
| 8.4375 | -19.7 | -14.1 |
| 8.5625 | -20.4 | -24.3 |
| 8.6875 | -35.6 | -42.8 |

From this exercise, we conclude that although some differences arise between periods, these differences are probably not the cause of the differences between the MDT and ADT velocities in the Panama Gulf when the "All-Sat" product is used.

Differences between the "2-Sat" and "All-Sat" ADT products

As we could not explain the differences between MDT and mean ADT currents in the Panama Gulf, based on time period differences, we assessed the second reviewer's suggestion. We downloaded daily ADT and geostrophic currents files from the "2-Sat" product ( https://data.marine.copernicus.eu/product/SEALEVEL_GLO_PHY_CLIMATE_L4_MY_008_0 57/description), for the study area and same period evaluated in the paper. We computed monthly mean values, which were used to compute mean behavior for the 1993-2012 period, using all the months, as well as means for March and November (Figure R2). We found that the 20 yr mean using all months for the 1993-2012 period in the Panama Bight showed a very similar behavior as the one seen in the MDT (1993-2012), as should be expected, and mentioned by the reviewer. Besides, in the March and November months (different seasons), the Panama Gulf currents were also westward. Bear in mind that both "All-Sat" and "2-Sat"products use the same MDT (CLS18– Mulet et al., (2021)) to calculate ADT.

Figure R2. ADT in centimeters (colorbar) and geostrophic currents.

[Figure]

[Figure]

[Figure]

**2Sat 20 yr mean**      **2Sat 20 yr March mean**      **2Sat 20 yr November mean**

This result surprised us, as we did not expect such differences between these two ADT products. We found that differences were only evident in the Panama Bight and not in the rest of the ETP. We speculate that differences might be related to: limitations of altimetry close to the coasts. Different number of altimeters available over time in the "All-Sat" product. MDT limitations at scales smaller than 100 km. Lack of observations in the Panama Gulf (CLS18 product also uses oceanographic in-situ measurements to calculate the MDT – Mulet (2021)). A combination of these possible reasons could also be the cause. Such limitations were probably the reason to calculate a specific MDT for the Mediterranean Sea (Rio, 2014 ), to overcome difficulties related to the Rossby radius, basin geometry and sharp coastal gradients in this region.

Due to the similarity of the "2-Sat" 1993-2012 mean with the MDT geostrophic currents, we determine that this product gives better results to assess the ocean dynamics in the ETP, but especially in the Panama Bight, where larger differences can be observed when compared to the "All-Sat" product. Besides, in the first version of the paper, we missed the paper from Chaigneau et al., (2006 ), in which a lagrangian study of the Panama Bight near surface circulation is done using drifters' data from 1979-2004. In their figure 4, seasonal circulation in the Panama Bight has a very good agreement with surface currents we show using the "2-Sat" product in the new version of the paper.

Furthermore, as mentioned in the "2-Sat" product description: "The processing focuses on the stability and homogeneity of the sea level record (based on a stable two-satellite constellation) and the product is dedicated to the monitoring of the sea level long-term evolution for climate applications and the analysis of Ocean/Climate indicators".

Therefore, in the new version of the paper we present the assessment of ADT and geostrophic currents in the study area using the more stable "2-Sat" product. The Data and methods section was updated accordingly, including a comment of the better results found with this product. Some of the paper's figures and tables were changed, as well as some of the results and discussions provided in the manuscript. The main difference is that with the "2-Sat" product we did not find a "reverse" circulation in the Panama Bight. The northerly Colombia Coastal Current is a permanent feature, as well as the westward circulation in the Panama Gulf. On the

contrary, the Panama Jet Surface Current is strong in the first third of the year, whereas it disappears during the boreal summer, when circulation is weaker and more variable in the Panama Bight. We include the Chaigneau et al., (2006) reference.

Another aspect of the Scientific Quality regards the dynamics. On line 54, the Panama wind jet is described as a continuation of the north trade winds from the Caribbean Sea to the Panama Bight. Kessler cites a number of papers that describe the jet as occurring during the very weak trade winds of boreal winter, driven by an atmospheric pressure difference between the Caribbean Sea and the ETP. In addition, the upwelling and downwelling is attributed by Kessler and others as due to the wind stress curl over the PB, whereas the authors simply attribute the upwelling as due to the wind jet, which could mean divergence at the coast.

We do not understand the comment "Kessler (2006) cites a number of papers that describe the jet as occurring during the very weak trade winds of boreal winter, ...". In Kessler (2006), section 4.2, discusses the annual cycle in the northeastern region. In the second paragraph of this section, Kessler references the work from Fiedler (2002), indicating that strong upwelling wind stress curl occurs in February – April (boreal winter). Similarly, during November-January, it blows strongly over the region. This is also shown in his Figure 9a. A recent paper from Bustos and Torres (2022) , assesses actual and projected seasonal changes in the Caribbean Low Level Jet, including some references related to this seasonality. Furthermore, Chelton et al. (2000), who is referenced by Kessler, indicates that the Papagayo and Panama jets are coupled to coherent variations of the trade winds extending from the Caribbean Sea to the ETP. This reference was included in the paper.

We clarified in the manuscript that the wind jet stress produces the wind stress curl, which forces the cyclonic circulation during boreal winter in the Panama Bight.

Other considerations for Scientific Quality include the methods and appropriate references. I'm not sure how much additional information is gained from the use of the SOM methodology. The mean from the SOM analysis should be similar to a simple mean of all of the 12 mean calendar months. Perhaps the SOM analysis discards any extreme outliers. I suggest a paragraph in the Introduction that could state the benefit of using the SOM analysis, i.e., what additional information is provided by its use. In the discussion on lines 395-415, if the pairs of the spatial neurons show the circulation in winter, then transitional months and then summer-autumn, those could be produced again by simple averages over those months.

SOM converts complex, non-linear statistical relationships among high-dimensional data into simple geometric relationships on a low-dimensional display. Similar to other data mining methods it uses statistical procedures to model  and handle noisy data. In other words, it compresses information while preserving the most important topological and metric relationships of the primary data. The essential point in the applicability of SOM is the topological character of the mapping: similar patterns are mapped in the nearby locations on the map performing a

topology that preserve mapping from the multi-dimensional input space onto map units so that relative distances between data points are preserved. Data points lying near each other in the input space will be mapped onto nearby map units. Thus, the average of SOM patterns of a physical variable  is not the mean of the 12 months calendar since there are some patterns that are included in the lattice as a ''transitional patterns'' to maintain the topology of the network. . Note that it has the ability to generalize; that is, the network can interpolate between previously encountered inputs. In this sense (among many others) SOM differs from the EOF where the sum of the patterns will give the original data.

In line 139, we indicate that SOM is used to confirm previous results shown in the ETP.  Besides, note that SOM is used in this work as a clustering tool of the high-dimensional input data used in the analysis. We believe this is important, as our approach, classifying the 324 months in three ENSO conditions, based on the SST anomalies of El Niño 3 and El Niño 1+2 has, as far as we know, not been previously used.

Regarding references, it's a matter of style, but other papers (Rodriguez-Rubio et al., 2003; Kessler, 2006; Devis-Marales et al., 2008) give credit to early papers, such as by Wooster, 1959; Wyrtki, 1965,1966; Stevenson, 1970.

We included the papers from Wooster, 1959 and Wyrtki, 1966, due to their importance in the early description of the ETP ocean dynamics. Besides, we also included the paper from Stevenson, 1970, due to his early contribution to the description of the circulation in the Panama Bight.

**Presentation Quality: Excellent (1)**
My main comment regarding the figures is to wonder why Figure A1 (the MDT results) is not included in the main body of the paper? These results are compared to those of the ADT, which are in the main body of the paper. Placing the MDT results elsewhere favors the depiction of the eastward velocities at the top of the Panama Bight, which may be an artifact of the ADT reference period (see above).

Thank you for this suggestion. We decided to move the MDT results to the main body of the paper as figure 2. Now this figure is coherent with ADT and geostrophic currents results shown in the other figures, obtained using the "2-Sat" product.

A complaint that is not the authors' fault is that the figures in the preprint that we are supposed to review are way too small to see clearly on a printed version (even with my magnifying glass). The obvious solution is to only look at the figures by displaying the pdf on a large screen and enlarging the figures greatly. This is what I did but it meant that I couldn't review the paper anywhere except in my office.
We apologize for this inconvenience. All of our figures are available upon request in a very good resolution.

*Specific Language/Grammar/Typos:*

1) Everywhere, starting on page 1: The authors use two language conventions to indicate the direction that winds and currents are moving: XXXXly and XXXXward. So currents moving toward the east are called both "easterly" and "eastward". The XXXXly form is confusing, since in the meteorological literature "easterly" means "from the east". I suggest using only the XXXXward (eastward), since that is not ambiguous. This occurs throughout the text.
Corrected.

2) Line 26: Perhaps change "limited" to "bordered".
Corrected.

3) Line 30: "reverse oceanic gyre" is unclear. "reversing oceanic gyre" would be more understandable.
Corrected.

4) Line 57: should be "as **a** response to"... Add the word "a".
Corrected.

5) Line 71: Eliminate the word "most". It should be "...one of the rainiest locations..."
Corrected.

6) Line 83: The wording is awkward. I suggest this wording or something like it: "as well as temporal sea level variability**, as represented by altimeter-derived** Sea Level Anomalies **(**SLA). Besides..."
Corrected.

7) Line 86: I suggest adding the word "determine: "...and **determine** if this forcing..."
Done.

8) Line 114: Eliminate the word "First" and the sentence starts: "Comparisons between..."
Done.

9) Line 131, 134: Awkward wording. The wording from line 134 would be better: "Anomalies are computed by subtracting **the 1993-2019 spatial mean from the individual monthly data** using all data..." On line 134, you do not need the word "respectively". Note that you are creating a different type of anomaly than the type of "Sea Level Anomaly" people think of with respect to altimeter data.
Corrected.

10) Line 151) Do you mean "**somewhat** subjective" ?
Yes. It has been corrected.

11) Line 171: Change "to" to "with". "Associated **with** ..."
Corrected.

12) Line 185: Add the word "**The**" at the beginning of the sentence.

Done.

13) Line 188 and others: Continue to change all of the XXXXly to XXXXward – here change "northerly" to "northward"...etc.
Done. We also check all these adverbs within all the entire document.

14) Line 190: Add the word "gyre" after "rotational". "...cyclonic rotational **gyre** closes..."
Done.

15) Line 192-3: The phrase "The mean temporal SOM..." is confusing. It might be better described as "The **temporal** mean of the SOM circulation ..." And at the end of the sentence, add the figure reference **(Figure A1c).**
This has been removed in the new version of the Ms.

16) Line 197: Change "to" to "of" "... east **of** 100...
Corrected.

17) Line 2019: "strength" should be "**strengthen**"
Corrected.

18) Many places (lines 207, 210, 237, 241, 279, etc.) The repeated use of the word "Besides" at the beginning of sentences is distracting and unneeded. In most of these sentences, the word "Besides" can simply be eliminated with no change in meaning. Sometimes, the word "also" can be added later in the sentence, as is done in line 279. If a connecting word is absolutely needed, other words might be better ("In addition", "Moreover", ..).
We thank the referee's comment. We have removed them from many parts of the document as suggested.

19) Line 237: "increase" should be "increases"
Corrected.

20) Line 239: Replace "by" with "of". "...is composed **of** the..."
Corrected.

21) Line 241: "eastern" should be "**eastward**"
Corrected.

22) Line 247: "reverse" could be "**reversing**"
Corrected.

23) Line 256: change tense from past to present, change "compared" to "**compare**"
Corrected.

24) Lines 267-268: Perhaps a minor point, but if "the first quarter of the year" is January-March, I believe there are only 7 months with El Niño positive conditions. There are 12 such months during January-April.
Corrected.

25) Line 287: The phrase "geostrophic currents seasonality" is awkward and could be replaced with "**the seasonality of the geostrophic currents**". If you keep the present wording, the word " **currents'** " needs an apostrophe.
We agree with the referee and we opt for his/her suggestion.

26) Line 289: The use of the word "de" is unusual but probably can be understood. Perhaps the phrase "underscored the fact that" is what is meant. Or just use the word "demonstrated".
We change "outlined"  by "demonstrated".

27) Lines 314-315: I do not understand the sentence that begins, "On the contrary,.." What is higher and less variable? The June-December ADT shown by the red line in Figure 5a seems more variable than the black line (normal) or the blue line (Niña)
We clarified this sentence.

28) Lines 375-376: I believe you have "eastward" and "westward" reversed. "The former" should be the SEC and that flow is "westward" at -34 cm/s. "the latter" should be the NECC and that flow is "eastward" with a value of +18.8 cm/s.
We apologize for this mistake which has been corrected.

28) Line 461: There is a typo: "ENOS" should be "ENSO"
Corrected throughout  the document.

---

## Author Comment (AC2)

**"Seasonal and ENSO-related ocean variability in the Panama Bight" by Rafael R. Torres, Estefanía Giraldo, Cristian Muñoz, Ana Caicedo, Ismael Hernández-Carrasco & Alejandro Orfila.**

We would like to thank the reviewer for their helpful and constructive comments to improve our paper. We have tried to respond to all of the comments and we hope that the paper has improved so as to now be acceptable for publication.

**Referee #2:**
The main goal of this study is to evaluate the ENSO effects on the interannual variability of the mean seasonal circulation patterns in the Panama Bight and Eastern Tropical Pacific region. For this purpose, different products were used to assess the hydrographic dynamics and physical variables in the region, such as, mean sea level, sea surface temperature, surface salinity and geostrophic currents. My overall appreciation is that this is a welcome addition to the literature of studies which evaluates the impact of climatic events on the physical dynamics, which have a direct impact on the biological and carbon cycles in the ocean. In my opinion, the data presented in the manuscript is highly valuable and the manuscript itself is mostly clearly presented and overall well-written, however, the methodology and discussion sections should be improved. Additionally, in general, the figures show the information in a clear manner, however, sometimes they are difficult to follow in connection with the main hypothesis of the manuscript. Please find below my explicit suggestions.

**Comments, questions or suggestions:**

**Methods:**

- Line 134: "Anomalies are computed by subtracting the 1993-2019 spatial mean using all data in the ETP (66.0 cm, 26.6 °C and 33.8 gr kg-1)." Does this indicate that only one average value was taken for the entire study area?. Why didn't the authors use spatial anomalies, i.e., considering pixel by pixel?. I ask this since not all of the Panama Bight region has the same conditions and it is a highly diverse region in physical terms.

All responses refer to lines and figures from the first version of the paper.

Lines 128-132 describes the methodology to compute ADT, SST and SSS monthly means under the three ENSO-related conditions used to assess spatial anomalies in the two main seasons (Figures 2, 3, 4 and Aux 2). Lines 133-138 describe the methodology to compute regionally averaged time series, showing anomalies for the 12 months of the year under the three ENOS-related conditions (Figures 5 and Aux 3). In both cases, monthly anomalies (under the three ENSO-related conditions), are obtained subtracting the multiannual monthly mean, a unique value computed regionally in the 1993-2019 period (e.g. 66.0 cm, 26.6 °C and 33.8 gr kg$^{-1}$ for the ETP). We followed this methodology in order to show, in the first case, spatial differences or gradients at seasonal and interannual (ENSO) time-scales. This is important in the case of ADT, as geostrophic currents result from these gradients. In the second case, this method allowed us to show the monthly variations of ocean properties (ADT, speed, SST, SSS) in a defined region (ETP and Panama Bight), as well as the interannual variations (related to ENSO) which affect the seasonality and the mean (interannual

shifts). Therefore, we consider that computing this method is suitable to show the seasonal circulation in the Panama Bight, and how this seasonality, as well as other ocean properties, are affected by ENSO, which are the main goals in the paper.

- Line 160: Why do the authors use El Niño 3 index here, if they had previously used the El Niño 1+2 index?. This is confusing to me. I believe it is important to discuss or clarify why the authors use two different types of ONI indices, mainly in relation to their use in the different subsequent analyses. Also, if the authors found similar ocurrence values with both indices (lines 117-123), why not use only one index?.

Lines 112-114, explains that Niño 1+2 is used as an indication of ENSO's local effect in the Panama Bight, as the SST region used for this index, partially covers the Cold Tongue, which affects the bight's air-ocean dynamics.

The El Niño-3 region is used for two purposes;  first, it allows us  to assess if results from the analysis stand with an ENSO index that indicates SST variations in the central equatorial Pacific. We think this is important, because we show that both indices have the same order in the frequency of occurrence (normal, La Niña, El Niño), but the percentages of occurrence are different (Lines 117-123). Furthermore, a comparison between panel f in Figures 5 and Aux 3 shows large differences in the months of occurrence of the ENOS conditions when Niño 1+2 and Niño 3 are used respectively (Lines 285-286).Second, as mentioned in Lines 159-162, the El Niño 3 is used to correlate SOM residual time series with ENOS. In this case, we use Niño 3 because SOM analysis is performed using all data from the ETP. Consequently, the six temporal SOM patterns and their corresponding time series (Figure 6) represent the variability of the entire ETP.

We believe  that the El Niño 3 index is more appropriate than the El Niño 1+2 to assess ENSO-related variability, when the area of study is the entire ETP, because of the SST areas used to compute these indices. In the former, the area is 5°N-5°S; 150°W-90°W, while in the latter the area is 0°-10°S; 90°W-80°W. In summary, , El Niño 3 SST region covers twice the region covered by El Niño 1+2 in the ETP defined in our study being the former  better to assess the relationship between the residual SOM time series with the ENOS. For completeness, it is appropriate to report also if the findings stand when the Niño 1+2 region is used as an additional result.

**Results and discussion:**

- It seems interesting to me that in section 3.1 the authors could discuss the reasons for the differences between SOM and MDT circulation in the Panama Bight, since this is the area of interest for this study. Ideally, a discussion is important in terms of validation of the SOM methodology.

Thank you for the comment. Based on a comment from Referee #1, we realized that the SOM mean circulation we used in the first version of the paper could be confusing. This is because the essential point in the applicability of SOM is the topological character of the mapping: similar patterns are mapped in the nearby locations on the map performing a topology that preserve mapping from the multi-dimensional input space onto map units so that relative distances between data points are preserved. Therefore, in the new version of the paper we remove all references to the SOM mean circulation. Please see the answer to

your next comment, where we indicate the SOM, MDT and ADT differences between the first and the new version of the paper.

- Again in section 3.2 the authors express differences by using ADT, MDT and SOM but do not discuss these differences in terms of methodology and/or which should be the most accurate methodology to be used. This discussion is also important, for example, to explain why the authors report differences related to ENSO events only considering the values obtained with the ADT product (Figure 5).

Thank you for the comment, as it opens an important opportunity to indicate main differences between the first version of the paper and the new one . Reviewer #1 asked us to verify the differences in  the circulation patterns that we found in the Panana Bight between MDT and ADT. The reviewer also gave us some suggestions for the  assessment. One recommendation was to compare the ADT mean circulation in the Bight with the MDT using the same period (1993-2012). Another recommendation was to verify an ADT product based on two satellites during the entire time series ("2-Sat"), instead of using the ADT product  using all available satellites ("All-Sat") , which was the one we used in the original paper.

Following this recommendation  we computed the annual mean circulation in the Panama Bight           using             the             ADT            "2-sat"               product (https://data.marine.copernicus.eu/product/SEALEVEL_GLO_PHY_CLIMATE_L4_MY_ 008_057/description) for the 1993-2012 period (Figure R3). Besides, we calculated the 20-year mean circulation for March and November, the same months used in the paper to assess seasonality.

Figure R3. "2-Sat" ADT in centimeters (colorbar) and geostrophic currents.

[Figure]

We found, as expected, a very consistent comparison between the mean ADT  and MDT circulation patterns, the latter shown in the paper's Figure A1-b. . Besides, in March and November (different seasons), the circulation patterns are also consistent  with the MDT, particularly in the Panama Gulf and the Colombia Coastal Current.

 Consequently, in this regard, the new version of the paper include the following changes:

- All ADT assessments are done with the "2-Sat" product, as it is dedicated to the monitoring of the sea level long-term evolution for climate applications.

- MDT figure (previously in the supplementary material) has been moved to the main text as figure 2.

- Figure of the mean SOM circulation has been removed.

- SOM analysis is performed with the "2-Sat" product for consistency.

- The manuscript has been accordingly being modified to account with the new results. Note that now, the MDT and ADT circulation patterns coincide in the ETP, including the Panama Bight, which was the main problem in the former version of the paper.

After we posted Referee's #1 response, we continue investigating the reasons of the ADT circulation differences between the two available products ("2-Sat" and "All-Sat"). Thus, we downloaded the latest version of the "All-Sat" product (https://data.marine.copernicus.eu/product/SEALEVEL_GLO_PHY_L4_MY_008_047/des cription), and computed the Panama Bight circulation pattern for the 1993-2012 period, as we did with the "2-Sat" product (Figure R4). Note that results are very similar in the two ADT products, what give us confidence that the inconsistency in the Panama Bight circulation, shown in the first version of the paper, is resolved in the new version.

Figure R4. "All-Sat" ADT in centimeters (colorbar) and geostrophic currents.

[Figure]

- Lines 275-276: What are the authors' criteria for limiting sea level ranges?

ADT ranges of 35 cm were defined based on the ADT anomalies variations observed in the normal ENSO conditions in Figure 2 (b and e) and Figure 3 (b and f).

One of the findings of the paper is that positive/negative ENSO conditions shift upward/downward the regional mean ADT without changing the circulation patterns. As geostrophic currents respond to ADT gradients and not to the mean level, we decided to

highlight the ADT gradients by maintaining the 35 cm range in these two figures. However, in order to account for the ENSO-related shifts in mean ADT, the color scale is also shifted +10/-5 cm for the positive/negative ENSO conditions, when compared to normal conditions. Thus, the reader can see that ADT gradients (circulation) do not change much due to ENSO, regardless of the regional mean shifts produced by this climatic pattern. This was explained in Lines 215-216 & 230-231 in the s' legend, as well as in Lines 272-275 in the Results section 3.3 (circulation variations related to ENSO). In the new version of the Manuscript , we complemented briefly the description of the sea level ranges, given in section 3.3.

- Lines 284-288 and 307-309: I recommend reviewing "VARIATION IN THE SURFACE CURRENTS IN THE PANAMA BIGHT DURING EL NIÑO AND LA NIÑA EVENTS FROM 1993 TO 2007 by Corredor-Acosta et al., 2011" where differences in the velocity of different geostrophic currents and cyclonic/anticyclonic circulation patterns were observed in relation to different ENSO events and neutral years. Perhaps the authors did not find differences because they took a regional average value to calculate the anomalies?

Thank you for mentioning this paper, which we missed in our bibliographic review. Corredor et al. (2011) reports statistical differences in the NECC, SEC intensity and in two other circulation systems, denominated Coastal Current and Anticyclonic eddy, shown in their Figure 1b (mean currents for September 1995 under La Niña conditions). These last two circulation patterns are not seen neither in the MDT nor in the ADT March and November averaged currents shown  in our study (Figure 3), regardless of the ENSO-related condition. Besides, we could not find references to these circulation patterns in the bibliographic review.

Corredor et al.  methodology is very different to the one that we follow in our study. They use the Oceanic Niño Index, which uses SST in the 3.4 region. Their results are limited to September to November, using four years representative of moderate El Niño, three years for moderate La Niña, and four neutral years, all in the 1993-2007 period. They use total currents, estimated as the sum of the Ekman currents and surface geostrophic currents. The latter are computed from the geostrophic relation using ADT, which has a limitation close to the Equator due to the small Coriolis effect. For this reason, they excluded the band between 0-1°N. The statistical analysis is performed for four sub-regions described in their Table 1, three of them south of 5°N.

On the contrary, we focus on the Panama Bight, and include the ETP to give context to our findings. We claim that seasonal circulation differences are stronger than ENOS related variations, giving evidence of the physical process based on SST and SSS variations, strongly related to air-sea interaction local processes (Panama and Choco wind surface jets). Besides, in the abstract we indicate that  "ENSO … climatic variability does not modify the seasonal circulation patterns in the Panama Bight". This finding is supported in three ways: (i)  the comparison in the spatial and temporal (both seasonal and interannual) circulation patterns shown for the ETP (Figure 2) and Panama Bight (Figure 3); (ii) , in the ETP and Panama Bight seasonality shown with regional averages (Figure Aux 3 and Figure 5 respectively) and (iii) the SOM analysis (Section 3.5).

Furthermore, our results are consistent with findings from Chaigneau et al. (2006), a study that we missed in the first version of our paper which has been included . In the updated Ms, , a similar seasonal circulation is shown based on 25 years of satellite-tracked drifters' trajectories in the Panama Bight.

For completeness, we included the Corredor et al. (2011) work , at the end of Section 3.3. We indicate that statistical differences in the currents  might exist in some areas. For this reason, we report small differences in the ENSO-related circulation patterns in both, the ETP and Panama Bight.

**Summary and final remarks:**

I suggest that this section be limited to the main results obtained according to the main hypothesis proposed in the manuscript.

Thank you for the suggestion. Indeed, in the updated version of the paper, the first paragraph of this section is changed to present the results of the main problem addressed with the paper in a more clear way. Consequently, we explain the observed seasonal circulation patterns in the Panama Bight, which includes a permanent Colombia Coastal Current and westward flow in the Panama Gulf, regardless of other strong seasonal circulation differences.

We prefer to present a "Summary and final remarks" section, instead of a "Conclusions" section because, in our point of view, many readers after revising the abstract, will  check this last section, in order to review the most relevant results of the work .

**Minor suggestions:**

- Please review some citations in the manuscript where compound surnames should be hyphenated, e.g., Rueda-Bayona; Rodríguez-Rubio; Devis-Morales.

Corrected

- Line 87: "….we also assess SST and Sea Surface Salinity (SSS) variability in the region in seasonal and interannual timescales." Please change "in seasonal" by "at seasonal".

Corrected

- Lines 152-153: Please specify what means "sufficient resolution and statistical accuracy"

This refers to the compromise between accuracy of the results and resolution of the structures obtained. In general,  a large number of neurons  produce a large number of small but compact clusters (records assigned to each cluster are quite similar). Small maps

produce less but more generalized clusters. A "right number of clusters' ' doesn't exist, especially in real world datasets. It all depends on the detail which one wants to examine the specific dataset.

- Line 344: I suggest specify the most relevant references rather than only "section 2".

Corrected, we included two references (Poveda and Mesa, 2000; Hastenrath and Lamb, 2004).

 References

Chaigneau, A., Abarca del Rio, R., and Colas, F.: Lagrangian study of the Panama Bight and surrounding regions, Journal of Geophysical Research: Oceans, 111, https://doi.org/10.1029/2006JC003530, 2006.

Corredor, A., Acosta, A., Gaspar, P., and Calmettes, B.: Variation in the surface currents in the Panama Bight during El NIño and La Niña Events from 1993 to 2007, Boletín de Investigaciones Marinas y Costeras, 40, https://doi.org/10.25268/bimc.invemar.2011.40.0.127, 2011.

Hastenrath, S. and Lamb, P. J.: Climate dynamics of atmosphere and ocean in the equatorial zone: a synthesis, International Journal of Climatology, 24, 1601–1612, https://doi.org/10.1002/joc.1086, 2004.

Poveda, G. and Mesa, O. J.: On the existence of Lloró (the rainiest locality on Earth): Enhanced ocean-land-atmosphere interaction by a low-level jet, Geophysical Research Letters, 27, 1675–1678, https://doi.org/10.1029/1999GL006091, 2000.

---

## Author Response (AR2)

Dear Editor,

We would like to thank the Reviewer as well as the Editor for the process of reviewing our Ms. We think that thanks to their support we ended with a much solid presentation.

Dr. Alejandro Orfila, on regards of all co-authors